# Generalized Bayesian Inference for Scientific Simulators via Amortized Cost Estimation

**Richard Gao**[*,1,2]
r.dg.gao@gmail.com

**Michael Deistler**[*,1,2]
michael.deistler@uni-tuebingen.de

**Jakob H. Macke**[1,2,3]
jakob.macke@uni-tuebingen.de

[1]Machine Learning in Science, Excellence Cluster Machine Learning, University of Tübingen
[2]Tübingen AI Center
[3]Department Empirical Inference, Max Planck Institute for Intelligent Systems
Tübingen, Germany
[*]Equal contributions.

## Abstract

Simulation-based inference (SBI) enables amortized Bayesian inference for simulators with implicit likelihoods. But when we are primarily interested in the quality of predictive simulations, or when the model cannot exactly reproduce the observed data (i.e., is misspecified), targeting the Bayesian posterior may be overly restrictive. Generalized Bayesian Inference (GBI) aims to robustify inference for (misspecified) simulator models, replacing the likelihood-function with a cost function that evaluates the goodness of parameters relative to data. However, GBI methods generally require running multiple simulations to estimate the cost function at each parameter value during inference, making the approach computationally infeasible for even moderately complex simulators. Here, we propose amortized cost estimation (ACE) for GBI to address this challenge: We train a neural network to approximate the cost function, which we define as the expected distance between simulations produced by a parameter and observed data. The trained network can then be used with MCMC to infer GBI posteriors for any observation without running additional simulations. We show that, on several benchmark tasks, ACE accurately predicts cost and provides predictive simulations that are closer to synthetic observations than other SBI methods, especially for misspecified simulators. Finally, we apply ACE to infer parameters of the Hodgkin-Huxley model given real intracellular recordings from the Allen Cell Types Database. ACE identifies better data-matching parameters while being an order of magnitude more simulation-efficient than a standard SBI method. In summary, ACE combines the strengths of SBI methods and GBI to perform robust and simulation-amortized inference for scientific simulators.

## 1 Introduction

Mechanistic models expressed as computer simulators are used in a wide range of scientific domains, from astronomy, geophysics, to neurobiology. The parameters of the simulator, $\boldsymbol{\theta}$, encode mechanisms of interest, and simulating different parameter values produces different outputs, i.e., $\text{sim}(\boldsymbol{\theta}_i) \to \mathbf{x}_i$, where each model-simulation $\mathbf{x}_i$ can be compared to experimentally observed data, $\mathbf{x}_o$. Using such simulators, we can quantitatively reason about the contribution of mechanisms behind experimental

37th Conference on Neural Information Processing Systems (NeurIPS 2023).

measurements. But to do so, a key objective is often to find all those parameter values that can produce simulations consistent with observed data.

One fruitful approach towards this goal is simulation-based inference (SBI) [1], which makes it possible to perform Bayesian inference on such models by interpreting simulator outputs as samples from an implicit likelihood [2], $\mathbf{x} \sim p(\mathbf{x}|\boldsymbol{\theta})$. Standard Bayesian inference targets the parameter posterior distribution given observed data, i.e., $p(\boldsymbol{\theta}|\mathbf{x}_o) = \frac{p(\mathbf{x}_o|\boldsymbol{\theta})p(\boldsymbol{\theta})}{p(\mathbf{x}_o)}$, where $p(\boldsymbol{\theta})$ captures prior knowledge and constraints over model parameters, and the likelihood function $p(\mathbf{x}_o|\boldsymbol{\theta})$ is evaluated as a function of $\boldsymbol{\theta}$ for a fixed $\mathbf{x}_o$. SBI methods can differ in whether they aim to approximate the likelihood [3, 4, 5, 6] or the posterior directly [7, 8, 9], and can be *amortized*, i.e., do not require new simulations and retraining for new data [10, 11]. In the end, each method provides samples from the posterior, which are all, in theory, capable of producing simulations that are *identical* to the observation we condition on. Furthermore, by definition, the posterior probability of drawing a sample scales as the product of its prior probability and, critically, the likelihood that this sample can produce a simulation that is *exactly equal* to the observation.

However, targeting the exact posterior may be overly restrictive. In many inference scenarios, modelers are primarily interested in obtaining a diverse collection of parameter values that can explain the observed data. This desire is also reflected in the common usage of posterior predictive checks, where seeing predictive simulations that resemble the data closely (in some specific aspects) is used to gauge the success of the inference process. In particular, it is often clear that the scientific model is only a coarse approximation to the data-generating process, and in some cases even cannot generate data-matching simulations, i.e., is misspecified [12]. For example, in the life-sciences, it is not uncommon to use idealized, theoretically motivated models with few parameters, and it would be unrealistic to expect that they *precisely* capture observations of highly complex biological systems. In such cases, or in cases where the model is fully deterministic, it is nonsensical to use the probability of exactly reproducing the data. In contrast, it would still be useful to find parameter values that produce simulations which are 'close enough', or as close as possible to the data. Therefore, instead of sampling parameters according to *how often* they produce simulations that match the data *exactly*, many use cases call for sampling parameters according to *how closely* their corresponding simulations reproduce the observed data.

**Generalized Bayesian Inference (GBI)** [13] offers a principled way to do so by replacing the (log) likelihood function with a cost function that simply scores a parameter given an observation, such as the expected distance between $\mathbf{x}_o$ and all possible simulations $\mathbf{x}$ produced by $\boldsymbol{\theta}_i$ (Fig. 1). Several recent works have leveraged this framework to perform inference for models with implicit or intractable likelihoods, especially to tackle model misspecification: Matsubara et al. [14] use a Stein Discrepancy as the cost function (which requires the evaluation of an unnormalized likelihood and multiple i.i.d. data samples), and Cherief-Abdellatif et al. and Dellaporta et al. [15, 16] use simulator samples to estimate maximum mean discrepancy and directly optimize over this cost function via stochastic gradient descent (which

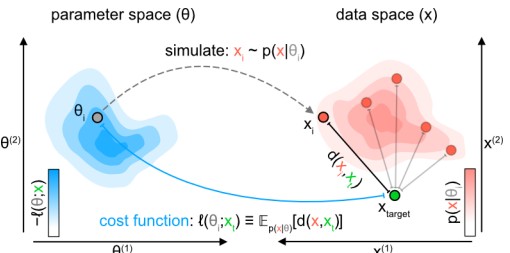

Figure 1: **Estimating cost from simulations.** Using the expected distance between simulated and target data as the cost function, GBI assigns high probability to parameter values that, on average, produce simulations that are close—but not necessarily equal—to the observation.

requires a differentiable simulator). More broadly, cost functions such as scoring rule estimators have been used to generalize approximate Bayesian computation (ABC) [17] and synthetic likelihood approaches [3, 18], where the Monte Carlo estimate requires (multiple) simulations from $p(\mathbf{x}|\boldsymbol{\theta})$.

Thus, existing GBI approaches for SBI either require many simulations to be run during MCMC sampling of the posterior (similar to classical ABC methods), or are limited to differentiable simulators. Moreover, performing inference for new observations requires re-running simulations, rendering such methods simulation-inefficient and expensive at inference-time, and ultimately impractical for scientific simulators with even moderate computational burden.

We here propose to perform GBI for scientific simulators with **amortized cost estimation (ACE)**, which inherits the flexibility of GBI but amortizes the overhead of simulations by training a neural

network to predict the cost function for any parameter-observation pair. We first outline the GBI formalism in Sec. 2, then introduce ACE in Sec. 3. In Sec. 4, we show that ACE provides GBI posterior predictive simulations that are close to synthetic observations for a variety of benchmark tasks, especially when the simulator is misspecified. We showcase its real-world applicability in Sec. 5: using experimental data from the Allen Cell Types Database, ACE successfully infers parameters of the Hodgkin-Huxley single-neuron simulator with superior predictive performance and an order of magnitude higher simulation efficiency compared to neural posterior estimation [10]. Finally, we discuss benefits and limitations of GBI and ACE, and related work (Sec. 6).

## 2   Background

To construct the GBI posterior, the likelihood, $p(\mathbf{x}_o|\boldsymbol{\theta})$, is replaced by a 'generalized likelihood function', $L(\boldsymbol{\theta}; \mathbf{x}_o)$, which does not need to be a probabilistic model of the data-generating process, as long as it can be evaluated for any pair of $\boldsymbol{\theta}$ and $\mathbf{x}_o$. Following the convention in Bissiri et al. [13], we define $L(\boldsymbol{\theta}; \mathbf{x}_o) \equiv e^{-\beta\ell(\boldsymbol{\theta}; \mathbf{x}_o)}$, where $\ell(\boldsymbol{\theta}; \mathbf{x}_o)$ is a cost function that encodes the quality of $\boldsymbol{\theta}$ relative to an observation $\mathbf{x}_o$, and $\beta$ is a scalar inverse temperature hyperparameter that controls how much the posterior weighs the cost relative to the prior. Thus, the GBI posterior can be written as

$$p(\boldsymbol{\theta}|\mathbf{x}_o) \propto e^{-\beta\ell(\boldsymbol{\theta}; \mathbf{x}_o)}p(\boldsymbol{\theta}). \tag{1}$$

As noted previously [13], if we define $\ell(\boldsymbol{\theta}; \mathbf{x}_o) \equiv -\log p(\mathbf{x}_o|\boldsymbol{\theta})$ (i.e., self-information) and $\beta = 1$, then we recover the standard Bayesian posterior, and "tempered" or "power" posterior for $\beta \neq 1$ [19, 20]. The advantage of GBI is that, instead of adhering strictly to the (implicit) likelihood, the user is allowed to choose arbitrary cost functions to rate the goodness of $\boldsymbol{\theta}$ relative to an observation $\mathbf{x}_o$, which is particularly useful when the simulator is misspecified. Previous works have referred to $\ell(\boldsymbol{\theta}; \mathbf{x}_o)$ as risk function [21], loss function [13], or (proper) scoring rules when they satisfy certain properties [22, 18] (further discussed in Section 6.1). Here we adopt 'cost' to avoid overloading the terms 'loss' and 'score' in the deep learning context.

## 3   Amortized Cost Estimation for GBI

### 3.1   Estimating cost function with neural networks

In this work, we consider cost functions that can be written as an expectation over the likelihood:

$$\ell(\boldsymbol{\theta}; \mathbf{x}_o) \equiv \mathbb{E}_{p(\mathbf{x}|\boldsymbol{\theta})}[d(\mathbf{x}, \mathbf{x}_o)] = \int_{\mathbf{x}} d(\mathbf{x}, \mathbf{x}_o)p(\mathbf{x}|\boldsymbol{\theta})d\mathbf{x}. \tag{2}$$

Many popular cost functions and scoring rules can be written in this form, including the average mean-squared error (MSE) [13], the maximum mean discrepancy (MMD$^2$) [23], and the energy score (ES) [22] (details in Appendix A2). While $\ell(\boldsymbol{\theta}; \mathbf{x}_o)$ can be estimated via Monte Carlo sampling, doing so in an SBI setting is simulation-inefficient and time-intensive, since the inference procedure must be repeated for every observation $\mathbf{x}_o$, and simulations must be run in real-time during MCMC sampling of the posterior. Furthermore, this does not take advantage of the structure in parameter-space or data-space around neighboring points that have been simulated.

We propose to overcome these limitations by training a regression neural network (NN) to learn $\ell(\boldsymbol{\theta}; \mathbf{x}_o)$. Our first insight is that cost functions of the form of Eq. 2 can be estimated from a dataset consisting of pairs of parameters and outputs—in particular, from *a single* simulation run per $\boldsymbol{\theta}$ for MSE, and finitely many simulation runs for MMD$^2$ and ES. We leverage the well-known property that NN regression converges to the conditional expectation of the data labels given the data: If we compute the distances $d(\mathbf{x}, \mathbf{x}_o)$ between a single observation $\mathbf{x}_o$ and every $\mathbf{x}$ in our dataset, then a neural network $f_\phi(\cdot)$ trained to predict the distances given parameters $\boldsymbol{\theta}$ will denoise the noisy distance labels $d(\mathbf{x}, \mathbf{x}_o)$ and converge onto the desired cost $f_\phi(\boldsymbol{\theta}) \to \ell(\boldsymbol{\theta}; \mathbf{x}_o) = \mathbb{E}_{p(\mathbf{x}|\boldsymbol{\theta})}[d(\mathbf{x}, \mathbf{x}_o)]$, approximating the cost of any $\boldsymbol{\theta}$ relative to $\mathbf{x}_o$ (see Appendix A3.1 for formal statement and proof).

### 3.2   Amortizing over observations

As outlined above, a regression NN will converge onto the cost function $\ell(\boldsymbol{\theta}; \mathbf{x}_o)$ for a particular observation $\mathbf{x}_o$. However, naively applying this procedure would require retraining of the network

for any new observation $\mathbf{x}_o$, which prevents application of this method in time-critical or high-throughput scenarios. We therefore propose to *amortize* cost estimation over a target distribution $p(\mathbf{x}_t)$. Specifically, a NN which receives as input a parameter $\boldsymbol{\theta}$ and an independently sampled target datapoint $\mathbf{x}_t$ will converge to $\ell(\boldsymbol{\theta}; \mathbf{x}_t)$ for all $\mathbf{x}_t$ on the support of the target distribution (Fig. 2a,b), enabling estimation of the cost function for any pair of $(\boldsymbol{\theta}, \mathbf{x}_t)$ (see Appendix A3.2 for formal statement and proof).

Naturally, we use the already simulated $\mathbf{x} \sim p(\mathbf{x})$ as target data during training, and therefore do not require further simulations. In order to have good accuracy on potentially misspecified observations, however, such datapoints should be within the support of the target distribution. Thus, in practice, we augment this target dataset with noisy simulations to broaden the support of $p(\mathbf{x}_t)$. Furthermore, if the set of observations (i.e., real data) is known upfront, they can also be appended to the target dataset during training. Lastly, to keep training efficient and avoid quadratic scaling in the number of simulations, we randomly subsample a small number of $\mathbf{x}_t$ per $\boldsymbol{\theta}$ in each training epoch (2 in our experiments), thus ensuring linear scaling as a function of simulation budget. Fig. 2a,b summarizes dataset construction and network training for ACE (details in Appendix A4.1).

### 3.3 Sampling from the generalized posterior

Given a trained cost estimation network $f_\phi(\cdot, \cdot)$, an observed datapoint $\mathbf{x}_o$, and a user-selected inverse temperature $\beta$, the generalized posterior probability (Eq. 1) can be computed up to proportionality for any $\boldsymbol{\theta}$: $p(\boldsymbol{\theta}|\mathbf{x}_o) \propto \exp(-\beta \cdot f_\phi(\boldsymbol{\theta}, \mathbf{x}_o))p(\boldsymbol{\theta})$ and, thus, this term can be sampled with MCMC (Fig. 2c). The entire algorithm is summarized in Algorithm 1.

---

**Algorithm 1:** Generalized Bayesian Inference with Amortized Cost Estimation (ACE)

---

**Inputs:** prior $p(\boldsymbol{\theta})$, simulator with implicit likelihood $p(\mathbf{x}|\boldsymbol{\theta})$, number of simulations $N$, feedforward NN $f_\phi$ with parameters $\phi$, NN learning rate $\eta$, distance function $d(\cdot, \cdot)$, noise level $\sigma$, number of noise-augmented samples $S$, inverse temperature $\beta$, number of target datapoints per $\boldsymbol{\theta}$ $N_{\text{target}}$, $K$ observations $\mathbf{x}_o^{(1,...,K)}$.
**Outputs:** $M$ samples from generalized posteriors given $K$ observations.

**Generate dataset:**
sample prior and simulate: $\boldsymbol{\theta}, \mathbf{x} \leftarrow \{\boldsymbol{\theta}_i \sim p(\boldsymbol{\theta}), \mathbf{x}_i \sim p(\mathbf{x}|\boldsymbol{\theta}_i)\}_{i:1...N}$
add noise and concatenate: $\mathbf{x}_{\text{target}} = [\mathbf{x}, \mathbf{x}_{1...S} + \boldsymbol{\epsilon}, \mathbf{x}_o], \boldsymbol{\epsilon} \sim \mathcal{N}(\mathbf{0}, \sigma^2 \mathbf{I})$

**Training:**
**while** *not converged* **do**
    **for** $(\boldsymbol{\theta}, \mathbf{x})$ *in batch* **do**
        $\mathbf{x}_t^{\text{used}} \leftarrow$ sample $N_{\text{target}}$ datapoints from $\mathbf{x}_{\text{target}}$
        **for** $\mathbf{x}_t$ *in* $\mathbf{x}_t^{used}$ **do**
            $\mathcal{L} \leftarrow \mathcal{L} + (f_\phi(\boldsymbol{\theta}, \mathbf{x}_t) - d(\mathbf{x}, \mathbf{x}_t))^2$
    $\phi \leftarrow \phi - \eta \cdot \text{Adam}(\nabla_\phi \mathcal{L})$ ;                  `// and reset L to zero`

**Sampling:**
**for** $k \in [1, ..., K]$ **do**
    Draw $M$ samples, with MCMC, from: $\exp(-\beta \cdot f_\phi(\boldsymbol{\theta}, \mathbf{x}_o^{(k)})) \, p(\boldsymbol{\theta})$

---

### 3.4 Considerations for choosing the value of $\beta$

We note that the choice of value for $\beta$ is an important decision, though this is an issue not only for ACE but GBI methods in general [13]. A good "baseline" value for $\beta$ is such that the average distance across a subset of the training data is scaled to be in the same range as the (log) prior probability, both of which can be computed on prior simulations. From there, increasing $\beta$ sacrifices sample diversity for predictive distance, and as $\beta$ approaches infinity, posterior samples converge onto the minimizer of the cost function. In practice, we recommend experimenting with a range of (roughly) log-spaced values since, as we show below, predictive sample quality tend to improve with increasing $\beta$.

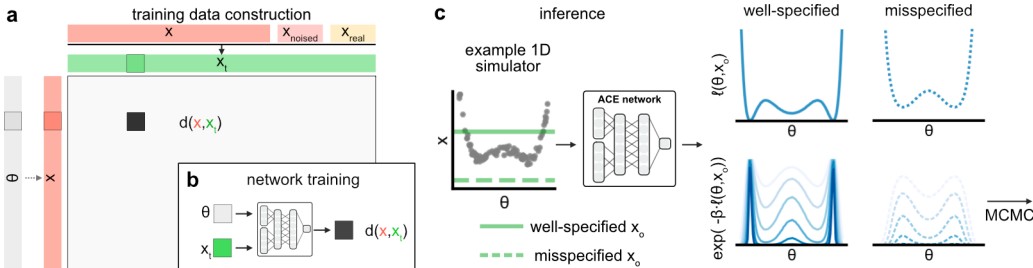

Figure 2: **Schematic of dataset construction, network training, and inference.** **(a-b)** The neural network is trained to predict the distance between pairs of $\mathbf{x}$ (red) and $\mathbf{x}_t$ (green), as a noisy sample of the cost function (i.e., expected distance) evaluated on $\boldsymbol{\theta}$ (grey) and $\mathbf{x}_t$. **(c)** At inference time, the trained ACE network predicts the cost for any parameter $\boldsymbol{\theta}$ given observation $\mathbf{x}_o$ (top row), which is used to evaluate the GBI posterior under different $\beta$ (bottom row, darker for larger $\beta$) for MCMC sampling without running additional simulations. The distance is well-defined and can be approximated even when the simulator is misspecified (dashed lines).

# 4 Benchmark experiments

## 4.1 Experiment setup

**Tasks**   We first evaluated ACE on four benchmark tasks (modified from Lueckmann et al. [24]) with a variety of parameter- and data-dimensionality, as well as choice of distance measure: (1) **Uniform 1D**: 1D $\boldsymbol{\theta}$ and $\mathbf{x}$, the simulator implements an even polynomial with uniform noise likelihood, uniform prior (Fig. 2c); (2) **2 Moons**: 2D $\boldsymbol{\theta}$ and $\mathbf{x}$, simulator produces a half-circle with constant mean radius and radially uniform noise of constant width, translated as a function of $\boldsymbol{\theta}$, uniform prior; (3) **Linear Gaussian**: 10D $\boldsymbol{\theta}$ and $\mathbf{x}$, Gaussian model with mean $\boldsymbol{\theta}$ and fixed covariance, Gaussian prior; (4) **Gaussian Mixture**: 2D $\boldsymbol{\theta}$ and $\mathbf{x}$, simulator returns five i.i.d. samples from a mixture of two Gaussians, both with mean $\boldsymbol{\theta}$, and fixed covariances, one with broader covariance than the other, uniform prior.

For the first three tasks, we use the mean-squared error between simulation and observation as the distance function. For the Gaussian Mixture task, we use maximum mean discrepancy ($\text{MMD}^2$) to measure the statistical distance between two sets of five i.i.d. samples. Importantly, for each of the four tasks, we can compute the integral in Eq. 2 either analytically or accurately capture it with quadrature over $\mathbf{x}$. Hence, we obtain the true cost $\ell(\boldsymbol{\theta}; \mathbf{x}_o)$ and subsequently the 'ground-truth' GBI posterior (with Eq. 1), and use that to draw, for each value of $\beta$ and $\mathbf{x}_o$, 5000 samples (GT-GBI, black in Fig. 3). See Appendix A4.2 for more detailed descriptions of tasks and distance functions.

**Training data**   For each task, we simulate 10,000 pairs of $(\boldsymbol{\theta}, \mathbf{x})$ and construct the target dataset as in Fig. 2a, with 100 additional noise-augmented targets and 20 synthetic observations—10 well-specified and 10 misspecified—for a total of 10120 $\mathbf{x}_t$ data points. Well-specified observations are additional prior predictive samples, while misspecified observations are created by moving prior predictive samples outside the boundaries defined by the minimum and maximum of 100,000 prior predictive simulations (e.g., by successively adding Gaussian noise). See Appendix A4.3 for details.

**Test data**   To evaluate inference performance, we use ACE to sample approximate GBI posteriors conditioned on 40 different synthetic observations, 20 of which were included in the target dataset $\mathbf{x}_t$, and 10 additional well-specified and misspecified observations which were not included in the target dataset. We emphasize that including observations in the target data is not a case of test data leakage, but represents a real use case where some experimental data which one wants to perform inference on are already available, while the network should also be amortized for unseen observations measured after training. Nevertheless, we report in Fig. 3 results for 'unseen' observations, i.e., not in the target dataset. Results are almost identical for those that were in the target dataset (Appendix A1). We drew 5000 posterior samples per observation, for 3 different $\beta$ values for each task.

**Metrics**   We are primarily interested in two aspects of performance: approximate posterior predictive distance and cost estimation accuracy. First, as motivated above, we want to find parameter

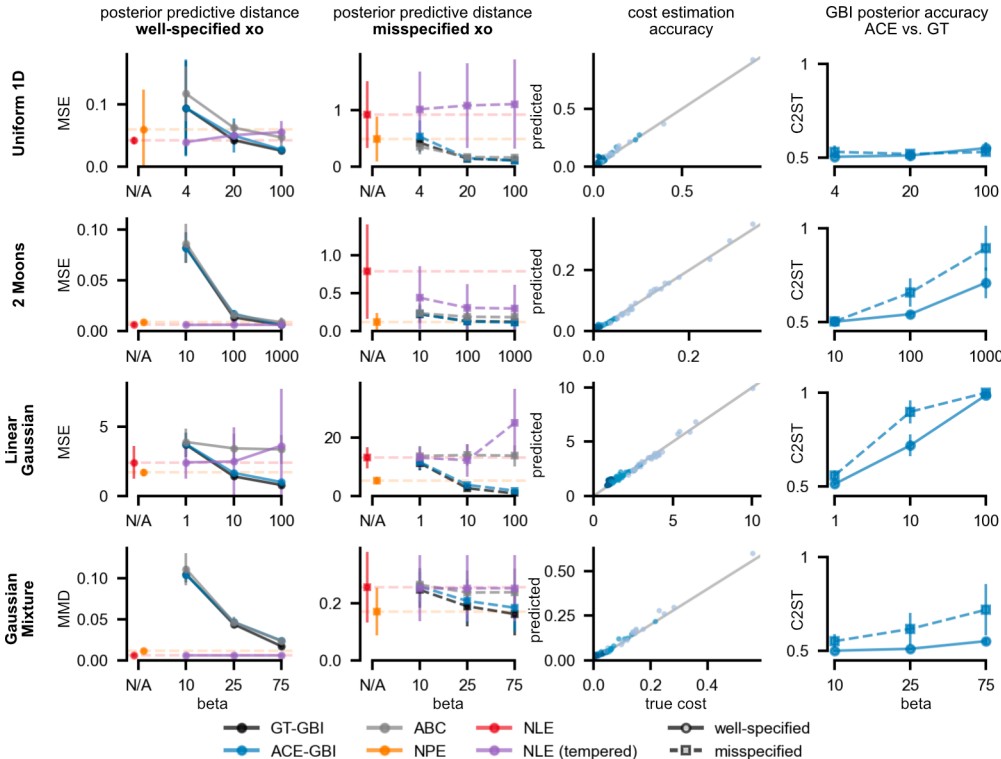

Figure 3: **Performance on benchmark tasks.** ACE obtains posterior samples with low average distance to observations, and accurately estimates cost function. **Rows:** results for each task. **Columns:** average predictive distance compared to SBI methods and GT (1st and 2nd), cost estimation accuracy evaluated on ACE posterior samples for different $\beta$ (lighter blue shades are lower values of $\beta$) (3rd), and C2ST accuracy relative to GT GBI posterior (4th, lower is better).

configurations which produce simulations that are as close as possible to the observation, as measured by the task-specific distance function. Therefore, we simulate using each of the 5000 ACE GBI posterior samples, and compute the average distance between predictive simulations and the observation. Mean and standard deviation are shown for well-specified and misspecified observations separately below (Fig. 3, 1st and 2nd columns). Second, we want to confirm that ACE accurately approximates $\ell(\boldsymbol{\theta}; \mathbf{x}_o)$, which is a prerequisite for correctly inferring the GBI posterior. Therefore, we compare the ACE-predicted and true cost across 5000 samples from each GBI posterior, as well as the classifier 2-sample test (C2ST, [25, 24]) score between the ACE approximate and ground-truth GBI posterior (Fig. 3, 3rd and 4th columns). Note that cost estimation accuracy can be evaluated for parameter values sampled in any way (e.g., from the prior), but here we evaluate accuracy as samples become more concentrated around good parameter values, i.e., from GBI posteriors with increasing $\beta$. We expect that these tasks become increasingly challenging with higher values of $\beta$, since these settings require the cost estimation network to be highly accurate in tiny regions of parameter-space.

**Other algorithms**   As a comparison against SBI methods that target the standard Bayesian posterior (but which nevertheless might produce good predictive samples), we also tested approximate Bayesian computation (ABC), neural posterior estimation (NPE, [7]), and neural likelihood estimation (NLE, [4, 26]) on the same tasks. NPE and NLE were trained on the same 10,000 simulations, and 5000 approximate posterior samples were obtained for each $\mathbf{x}_o$. We used the amortized (single-round) variants of both as a fair comparison against ACE. Additionally, to rule out the possibility that ACE is simply benefiting from the additional inverse temperature hyperparameter, we implemented a "tempered" version of NLE by scaling the NLE-approximated log-likelihood term (by the same $\beta$s) during MCMC sampling. For ABC, we used the 10,000 training samples as a reference set, from which 50 were drawn as posterior samples with probability scaling inversely with the distance between their corresponding simulation and the observation, i.e., ABC with acceptance kernel [27].

## 4.2 Benchmark results

Overall, we see that for well-specified $\mathbf{x}_o$ (i.e., observations for which the simulator is well-specified), ACE obtains GBI posterior samples that achieve low average posterior predictive simulation distance across all four tasks, especially at high values of $\beta$ (Fig. 3, 1st column). In comparison, ABC is worse for the Linear Gaussian task (which has a higher parameter dimensionality than all other tasks), whereas NPE, NLE, and tempered NLE achieve similarly low posterior predictive distances.

On misspecified observations, across all tasks and simulation-budgets (with the exception of Gaussian mixture on 10k simulations) we see that ACE achieves lower or equally low average posterior predictive simulation distance as neural SBI methods, even at moderate values of $\beta$ (Fig. 3, 2nd column, Figs. A4, A5). This is in line with our intuition that ACE returns a valid and accurate cost even if the simulator is incapable of producing data anywhere near the observation, while Bayesian likelihood and posterior probabilities estimated by NLE and NPE are in these cases nonsensical [28, 29, 30, 31]. Furthermore, simply concentrating the approximate Bayesian posterior via tempering does not lead to more competitive performance than ACE on such misspecified observations, and is sometimes even detrimental (e.g., tempered NLE at high $\beta$ on Uniform 1D and Linear Gaussian tasks). Therefore, we see that ACE can perform valid inference for a broad range of simulators, obtaining a distribution of posterior samples with predictive simulations close to observations, and is automatically robust against model-misspecification as a result of directly targeting the cost function.

For both well-specified and misspecified observations, ACE-GBI samples achieve posterior predictive distance very close to ground-truth (GT)-GBI samples, at all values of $\beta$ (Fig. 3, 1st and 2nd column), suggesting that ACE is able to accurately predict the expected distance. Indeed, especially for low to moderate values of $\beta$, the ACE-predicted cost closely matches the true cost (Fig. 3, 3rd column, light blue for well-specified $\mathbf{x}_o$, Fig. A2 for misspecified). For higher values of $\beta$, ACE-predicted cost is still similar to true cost, although the error is, as expected, larger for very large $\beta$ (Fig. 3, 3rd column, dark blue).

As a result, highly concentrated generalized posteriors are estimated with larger (relative) discrepancies, which is reflected in the classifier 2-sample score (C2ST) between ACE and GT GBI posteriors (Fig. 3, 4th column): ACE posterior samples are indistinguishable from GT samples at low $\beta$, even for the 10D Linear Gaussian task, but becomes less accurate with increasing $\beta$. Nevertheless, predictive simulation distance dramatically increases with $\beta$ even when ACE is less accurate, suggesting that sampling to explicitly minimize a cost function which targets parameters with data-similar simulations is a productive goal. Relative performance results across algorithms are qualitatively similar when using a training simulation budget of 200 (Fig. A4) and 1000 (Fig. A5), but ABC required a sufficiently high simulation budget and performed poorly for 1000 training simulations or less.

## 5 Hodgkin-Huxley inference from Allen Cell Types Database recordings

Finally, we applied ACE to a commonly used scientific simulator and real data: we used a single-compartment Hodgkin-Huxley (HH) simulator from neuroscience and aimed to infer eight parameters of the simulator given electrophysiological recordings from the Allen Cell Types Database [32, 33, 34]. While this simulator can generate a broad range of voltage traces, it is still a crude approximation to *real* neurons: it models only a subset of ion channels, it ignores the spatial structure of neurons, and it ignores many intracellular mechanisms [35]. It has been demonstrated that parameters of the HH-model given *synthetic* recordings can be efficiently estimated with standard NPE [10], but estimating parameters given *experimental* recordings has been challenging [36] and has required ad-hoc changes to the inference procedure (e.g., Bernaerts et al. [37] added noise to the summary statistics, and Gonçalves et al. [10] used a custom multi-round scheme with a particular choice of density estimator). We will demonstrate that ACE can successfully perform simulation-amortized inference given experimental recordings from the Allen Cell Types Database (Fig. 4a).

We trained NPE and ACE given 100K prior-sampled simulations (details in Appendix A4.4). After training, ACE accurately predicts the true cost of parameters given experimental observations (Fig. 4b). We then used slice sampling to draw samples from the GBI posterior for three different values of $\beta = \{25, 50, 100\}$ and for ten observations from the Allen Cell Types Database. Interestingly, the marginal distributions between NPE and ACE posteriors are very similar, especially for rather low values of $\beta$ (Fig. 4c, cornerplot in Appendix Fig. A7). The quality of posterior predictive samples, however, strongly differs between NPE and ACE: across the ten observations from the Allen

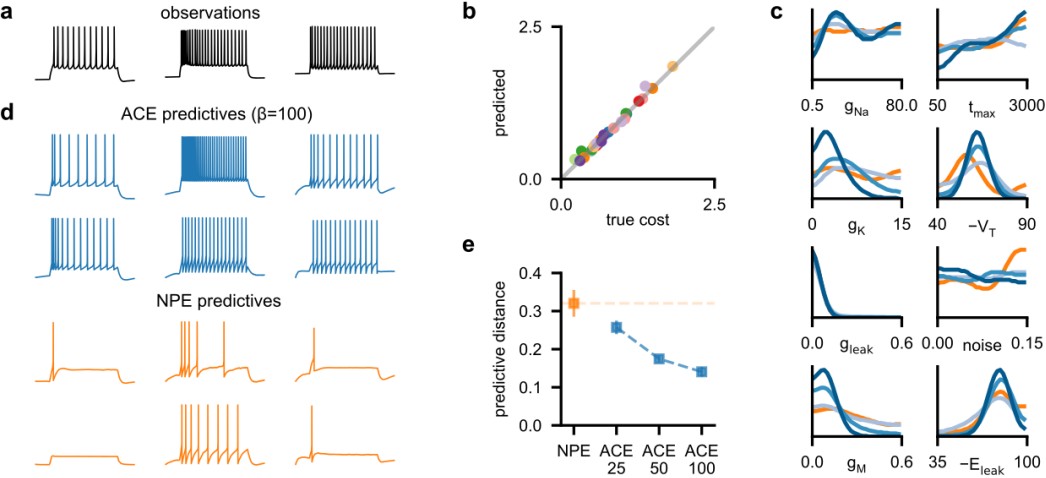

Figure 4: **Applicaton of ACE to Allen data. (a)** Three observations from the Allen Cell Types Database. **(b)** True cost (evaluated as Monte-Carlo average over 10 simulations) per $\theta$ vs ACE-predicted cost. Colors are different observations. **(c)** Marginals of posterior distributions for NPE (orange) and ACE (shades of blue. Light blue: $\beta = 25$, medium blue: $\beta = 50$, dark blue: $\beta = 100$). **(d)** Top: Two GBI predictive samples for each observation. Bottom: Two NPE predictive samples. Additional samples in Appendix A8-A11. **(e)** Average predictive distance to observation for NPE and ACE with $\beta = \{25, 50, 100\}$.

Cell Types database, only 35.6% of NPE posterior predictives produced more than five spikes (all observations have at least 12 spikes), whereas the ACE posterior predictives closely match the data, even for low values of $\beta$ (Fig. 4d, samples for all observations and all $\beta$ values in Figs. A8,A9,A10 and for NPE in Fig. A11. 66% ($\beta = 25$), 87% ($\beta = 50$), 96% ($\beta = 100$) of samples have more than five spikes). Indeed, across all ten observations, the average posterior predictive distance of ACE was significantly smaller than that of NPE, and for large values of $\beta$ the distance is even less than half (Fig. 4e). Finally, for rejection-ABC, only the top 35 samples (out of the full training budget of 100K simulations) had a distance that is less than the *average* posterior predictive distance achieved by ACE.

To investigate these differences between NPE and ACE, we also evaluated NPE posterior predictive performance on synthetic data (prior predictives) and found that it had an average predictive distance of 0.189, which roughly matches the performance of ACE on the experimental observations (0.174 for $\beta$=50). This suggest that, in line with previous results [37], NPE indeed struggles with *experimental* observations, for which the simulator is inevitably imperfect. We then trained NPE with 10 times more simulations (1M in total). With this increased simulation budget, NPE performed significantly better than with 100K simulations, but still produced poorer predictive samples than ACE trained with 100K simulations (for $\beta = \{50, 100\}$), although the marginals were similar between NPE (1M) and ACE (100K) (Fig. A6, samples for all observations in Appendix Fig. A12).

Overall, these results demonstrate that ACE can successfully be applied to real-world simulators on which vanilla NPE fails. On the Hodgkin-Huxley simulator, ACE generates samples with improved predictive accuracy despite an order of magnitude fewer simulations and despite the marginal distributions being similar to those of NPE.

## 6 Discussion

We presented ACE, a method to perform distance-aware inference for scientific simulators within the Generalized Bayesian Inference (GBI) framework. Contrary to 'standard' simulation-based inference (SBI), our method does not target the Bayesian posterior, but replaces the likelihood function with a cost function. For real-world simulators, doing so can provide practical advantages over standard Bayesian inference:

First, the likelihood function quantifies the probability that a parameter generates data which *exactly* matches the data. However, in cases where the model is a rather crude approximation to the real system being studied, scientists might well want to include parameters that can generate data that is sufficiently close (but not necessarily identical) in subsequent analyses. Our method makes this possible, and is advantageous over other GBI-based methods since it is amortized over observations and the inverse temperature $\beta$. Second, many simulators are formulated as noise-free models, and it can be hard to define appropriate stochastic extensions (e.g., [38]). In these cases, the likelihood function is ill-defined and, in practice, this setting would require 'standard' SBI methods, whose density estimators are generally built to model continuous distributions, to model discrete jumps in the posterior density. In contrast, our method can systematically and easily deal with noise-free simulators, and in such situations more closely resembles parameter-fitting algorithms. Lastly, standard Bayesian inference is challenging when the model is misspecified, and the performance of neural network-based SBI methods can suffer drastically in this scenario [30].

## 6.1 Related work

**GBI for Approximate Bayesian Computation** Several studies have proposed methods that perform GBI on simulators with either an implicit (i.e., simulation-based) likelihood or an unnormalized likelihood. Wilkinson et al. [39] argued that rejection-ABC performs exact inference for a modified model (namely, one that appends an additive uniform error) instead of approximate inference for the original model. Furthermore, ABC with arbitrary probabilistic acceptance kernels can also be interpreted as having different error models, and Schmon et al. [17] integrate this view to introduce generalized posteriors for ABC, allowing the user to replace the hard-threshold kernel (i.e., $\epsilon$-ball of acceptance) with an arbitrary loss function that measures the discrepancy between $\mathbf{x}$ and $\mathbf{x}_o$ for MCMC-sampling of the approximate generalized posterior.

Other recent GBI methods require a differentiable simulator [16, 15] or build tractable cost functions that can be sampled with MCMC [14, 18], but this still requires running simulations *at inference time* (i.e., during MCMC) and does not amortize the cost of simulations and does not reuse already simulated datapoints.

Finally, Bayesian Optimization for Likelihood-free Inference (BOLFI, [40]) and error-guided LFI-MCMC [41] are not cast as generalized Bayesian inference approaches, but are related to ACE. Similarly as in ACE, they train models (for BOLFI, a Gaussian process and, for error-guided LFI-MCMC, a classifier) to estimate the discrepancy between observation and simulation. In BOLFI, the estimator is then used to iteratively select new locations at which to simulate. However, contrary to ACE, neither of these two methods amortizes the cost of simulations over observations.

**Misspecification-aware SBI** Several other methods have been proposed to overcome the problem of misspecification in SBI: For example, Bernaerts et al. [37] add noise to the summary statistics in the training data, Ward et al. [42] use MCMC to make the misspecified data well-specified, and Kelly et al. [43] introduce auxiliary variables to shift the (misspecified) observation towards being well-specified. All of these methods, however, maintain that the inference result should be an 'as close as possible' version of the posterior distribution. Contrary to that, our method does *not* aim to obtain the Bayesian posterior distribution (which, for misspecified models, can often be nonsensical or even undefined if the evidence is zero), but is specifically targeted towards parameter regions that are a specified distance from the observation. More broadly, recent advances in uncertainty quantification in deep neural networks, where standard mean-predicting regression networks are supplemented with a uncertainty- or variance-predicting network [44, 45], may serve to further connect loss-minimizing deep learning with (misspecification-aware) SBI.

## 6.2 Limitations

While our method amortizes the cost of simulations and of training, it still requires another method to sample from the posterior distribution. We used multi-chain slice-sampling [46] for efficiency, but any other MCMC algorithm, as well as variational inference, could also be employed [47, 48]. While sampling incurs an additional cost, this cost is generally small in comparison to potentially expensive simulations.

In addition, our method can perform inference for distance functions which can be written as expectations over the likelihood. As we demonstrated, this applies to many popular and widely used distances. Our method can, however, not be applied to arbitrary distance functions (e.g., the minimum distance between all simulator samples and the observation). While the distances we investigated here are certainly useful to practioners, they do not necessarily fulfill the criterion of being 'proper' scoring rules [22, 18]. Furthermore, we note that the cost functions considered here by default give rise to unnormalized generalized likelihoods. Therefore, depending on whether the user aims to approximate the generalized posterior given the normalized or unnormalized likelihood, different MCMC schemes should be used in conjunction with ACE (standard MCMC vs. doubly-intractable MCMC, e.g., the Exchange algorithm [49]).

Compared to 'standard' SBI, GBI introduces an additional hyperparameter to the inference procedure, the inverse temperature $\beta$. This hyperparameter has to be set by the user and its choice strongly affects inference behaviour: low values of $\beta$ will include regions of parameter-space whose data do not necessarily match the observation closely, whereas high values of $\beta$ constrain the parameters to only the best-fitting parameter values. While we provide heuristics for selecting $\beta$, we acknowledge the inconvenience of an additional hyperparameter. However, our method is amortized over $\beta$, which makes exploration of different $\beta$ values possible, and which could simplify automated methods for setting $\beta$, similar to works where $\beta$ is taken as the exponent of the likelihood function [50].

Finally, as with any method leveraging deep neural networks, including neural density estimator-based SBI methods (such as NPE), sensitivity to the number of training samples and the dimensionality of the task should always be considered. As we demonstrate above, increasing simulation budget improves the performance of any algorithm, and a reasonable number of training simulations yielded improved performance on a real-world neuroscience application, while the amortization property shifts the cost of simulations up front. In addition, we consider tasks up to 10 dimensions here, as most existing SBI methods have been benchmarked and shown to perform adequately on such tasks [24], though it remains to be seen how ACE can extend to higher dimensional parameter-space and data-space and whether embedding networks will be similarly helpful.

# 7 Conclusion

We presented a method that performs generalized Bayesian inference with amortized cost estimation. Our method produces good predictive samples on several benchmark tasks, especially in the case of misspecified observations, and we showed that it allows amortized parameter estimation of Hodgkin-Huxley models given experimental recordings from the Allen Cell Types Database.

# 8 Acknowledgements

RG is supported by the European Union's Horizon 2020 research and innovation program under the Marie Skłodowska-Curie grant agreement No. 101030918 (AutoMIND). MD is supported by the International Max Planck Research School for Intelligent Systems (IMPRS-IS). The authors are funded by the Machine Learning Cluster of Excellence, EXC number 2064/1–390727645. This work was supported by the Tübingen AI Center (Agile Research Funds), the German Federal Ministry of Education and Research (BMBF): Tübingen AI Center, FKZ: 01IS18039A, and the German Research Foundation (DFG): SFB 1233, Robust Vision: Inference Principles and Neural Mechanisms, project number: 276693517.

We would like to thank Jan Boelts, Janne Lappalainen, and Auguste Schulz for feedback on the manuscript, Julius Vetter for feedback and discussion on proper scoring rules, Poornima Ramesh and Mackelab members for extensive discussions throughout the project, as well as Francois-Xavier Briol for suggestions on sampling doubly-intractable posteriors, and the reviewers for their constructive comments on the readability of the manuscript and suggestions for additional analyses.

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

# Appendix

## A1 Software and Data

We used PyTorch for all neural networks [51] and hydra to track all configurations [52]. Code to reproduce results is available at `https://github.com/mackelab/neuralgbi`.

## A2 Common cost functions as expectations of the likelihood

As described in Sec. 3, our framework includes all cost functions which can be written as expectations over the likelihood: $\ell(\boldsymbol{\theta}; \mathbf{x}_o) = \mathbb{E}_{p(\mathbf{x}|\boldsymbol{\theta})}[d(\mathbf{x}, \mathbf{x}_o)]$. Below, we demonstrate that the average mean-squared error (MSE), the maximum mean discrepancy (MMD$^2$) [23], and the energy score [22] fall into this category.

**Average Mean-squared error**    The average mean-squared error between samples from the likelihood $p(\mathbf{x}|\boldsymbol{\theta})$ and an observation $\mathbf{x}_o$ is defined as:

$$\text{MSE}(p(\mathbf{x}|\boldsymbol{\theta}), \mathbf{x}_o) = \mathbb{E}_{\mathbf{x} \sim p(\mathbf{x}|\boldsymbol{\theta})}[||\mathbf{x} - \mathbf{x}_o||^2]$$

which is trivially of the form required for our framework.

**Maximum mean discrepancy**    Throughout this paper, we use the maximum mean discrepancy *squared* (MMD$^2$) [23]. The MMD$^2$ between the likelihood $p(\mathbf{x}|\boldsymbol{\theta})$ and the data distribution $p(\mathbf{x}_o)$ is defined as:

$$\text{MMD}^2(p(\mathbf{x}|\boldsymbol{\theta}), p(\mathbf{x}_o)) =$$
$$\mathbb{E}_{\mathbf{x},\mathbf{x}' \sim p(\mathbf{x}|\boldsymbol{\theta})}[k(\mathbf{x}, \mathbf{x}')] + \mathbb{E}_{\mathbf{x}_o,\mathbf{x}'_o \sim p(\mathbf{x}_o)}[k(\mathbf{x}_o, \mathbf{x}'_o)] + \mathbb{E}_{\mathbf{x} \sim p(\mathbf{x}|\boldsymbol{\theta}), \mathbf{x}_o \sim p(\mathbf{x}_o)}[k(\mathbf{x}, \mathbf{x}_o)]$$

Assume we are given $K$ iid samples as observations $\mathbf{x}_o^{(1,\dots,K)}$. Thus, the MMD$^2$ becomes:

$$\text{MMD}^2(p(\mathbf{x}|\boldsymbol{\theta}), p(\mathbf{x}_o)) =$$
$$\mathbb{E}_{\mathbf{x},\mathbf{x}' \sim p(\mathbf{x}|\boldsymbol{\theta})}[k(\mathbf{x}, \mathbf{x}')] + \frac{1}{K^2} \sum_{i=1}^{K} \sum_{j=1}^{K} k(\mathbf{x}_o^i, \mathbf{x}_o^j) + \mathbb{E}_{\mathbf{x} \sim p(\mathbf{x}|\boldsymbol{\theta})} \frac{1}{K} \sum_i^K k(\mathbf{x}, \mathbf{x}_o^i)$$

which can be written as a single expectation over $p(\mathbf{x}|\boldsymbol{\theta})$

$$\text{MMD}^2(p(\mathbf{x}|\boldsymbol{\theta}), p(\mathbf{x}_o)) =$$
$$\mathbb{E}_{\mathbf{x} \sim p(\mathbf{x}|\boldsymbol{\theta})} \Big[ \mathbb{E}_{\mathbf{x}' \sim p(\mathbf{x}|\boldsymbol{\theta})}[k(\mathbf{x}, \mathbf{x}')] + \frac{1}{K^2} \sum_{i=1}^{K} \sum_{j=1}^{K} k(\mathbf{x}_o^i, \mathbf{x}_o^j) + \frac{1}{K} \sum_i^K k(\mathbf{x}, \mathbf{x}_o^i) \Big]$$

and follows the form required for our framework.

**Energy score**    Following Gneiting et al. [22], the (negative, in order to fit our notation) energy score (ES) is defined as

$$\text{ES}(p(\mathbf{x}|\boldsymbol{\theta})|\mathbf{x}_o) = -\frac{1}{2} \mathbb{E}_{\mathbf{x},\mathbf{x}' \sim p(\mathbf{x}|\boldsymbol{\theta})}[||\mathbf{x} - \mathbf{x}'||^\beta] + \mathbb{E}_{\mathbf{x} \sim p(\mathbf{x}|\boldsymbol{\theta})}[||\mathbf{x} - \mathbf{x}_o||^\beta].$$

This can straight-forwardly be written as a single expectation over the likelihood

$$\text{ES}(p(\mathbf{x}|\boldsymbol{\theta})|\mathbf{x}_o) = \mathbb{E}_{\mathbf{x} \sim p(\mathbf{x}|\boldsymbol{\theta})} [\frac{1}{2} \mathbb{E}_{\mathbf{x}' \sim p(\mathbf{x}|\boldsymbol{\theta})}[||\mathbf{x} - \mathbf{x}'||^\beta] - ||\mathbf{x} - \mathbf{x}_o||^\beta]$$

and, thus, falls within our framework.

## A3 Convergence proofs

The proofs closely follow standard proofs that regression converges to the conditional expectation.

### A3.1 Proposition 1 and proof

**Proposition 1.** *Let $p(\boldsymbol{\theta}, \mathbf{x})$ be the joint distribution over parameters and data. Let $\ell(\boldsymbol{\theta}; \mathbf{x}_o)$ be a cost function that can be written as $\ell(\boldsymbol{\theta}; \mathbf{x}_o) = \mathbb{E}_{p(\mathbf{x}|\boldsymbol{\theta})}[d(\mathbf{x}, \mathbf{x}_o)] = \int_{\mathbf{x}} d(\mathbf{x}, \mathbf{x}_o) p(\mathbf{x}|\boldsymbol{\theta}) d\mathbf{x}$ and let $f_\phi(\cdot)$ be a function parameterized by $\phi$. Then, the loss function $\mathcal{L} = \mathbb{E}_{p(\boldsymbol{\theta}, \mathbf{x})}[(f_\phi(\boldsymbol{\theta}) - d(\mathbf{x}, \mathbf{x}_o))^2]$ is minimized if and only if, for all $\boldsymbol{\theta} \in supp(p(\boldsymbol{\theta}))$, $f_\phi(\boldsymbol{\theta}) = \mathbb{E}_{p(\mathbf{x}|\boldsymbol{\theta})}[d(\mathbf{x}, \mathbf{x}_o)]$.*

*Proof.* We aim to prove that
$$\mathbb{E}_{\theta, \mathbf{x} \sim p(\boldsymbol{\theta}, \mathbf{x})}[(d(\mathbf{x}, \mathbf{x}_o) - g(\boldsymbol{\theta}))^2] \geq \mathbb{E}_{\theta, \mathbf{x} \sim p(\boldsymbol{\theta}, \mathbf{x})}[(d(\mathbf{x}, \mathbf{x}_o) - \mathbb{E}_{\mathbf{x}' \sim p(\mathbf{x}|\boldsymbol{\theta})}[d(\mathbf{x}', \mathbf{x}_o)])^2]$$
for every function $g(\boldsymbol{\theta})$. We begin by rearranging terms:
$$\mathbb{E}_{\theta, \mathbf{x} \sim p(\boldsymbol{\theta}, \mathbf{x})}[(d(\mathbf{x}, \mathbf{x}_o) - g(\boldsymbol{\theta}))^2] =$$
$$\mathbb{E}_{\theta, \mathbf{x} \sim p(\boldsymbol{\theta}, \mathbf{x})}[(d(\mathbf{x}, \mathbf{x}_o) - \mathbb{E}_{\mathbf{x}' \sim p(\mathbf{x}|\boldsymbol{\theta})}[d(\mathbf{x}', \mathbf{x}_o)] + \mathbb{E}_{\mathbf{x}' \sim p(\mathbf{x}|\boldsymbol{\theta})}[d(\mathbf{x}', \mathbf{x}_o)] - g(\boldsymbol{\theta}))^2] =$$
$$\mathbb{E}_{\theta, \mathbf{x} \sim p(\boldsymbol{\theta}, \mathbf{x})}[(d(\mathbf{x}, \mathbf{x}_o) - \mathbb{E}_{\mathbf{x}' \sim p(\mathbf{x}|\boldsymbol{\theta})}[d(\mathbf{x}', \mathbf{x}_o)])^2 + (\mathbb{E}_{\mathbf{x}' \sim p(\mathbf{x}|\boldsymbol{\theta})}[d(\mathbf{x}', \mathbf{x}_o)] - g(\boldsymbol{\theta}))^2] + X$$
with
$$X = \mathbb{E}_{\theta, \mathbf{x} \sim p(\boldsymbol{\theta}, \mathbf{x})}[(d(\mathbf{x}, \mathbf{x}_o) - \mathbb{E}_{\mathbf{x}' \sim p(\mathbf{x}|\boldsymbol{\theta})}[d(\mathbf{x}', \mathbf{x}_o)])(\mathbb{E}_{\mathbf{x}' \sim p(\mathbf{x}|\boldsymbol{\theta})}[d(\mathbf{x}', \mathbf{x}_o)] - g(\boldsymbol{\theta}))].$$
By the law of iterated expectations, one can show that $X = 0$:
$$X = \mathbb{E}_{\theta' \sim p(\boldsymbol{\theta})}\mathbb{E}_{\theta \sim p(\boldsymbol{\theta})}\mathbb{E}_{\mathbf{x} \sim p(\mathbf{x}|\boldsymbol{\theta})}[(d(\mathbf{x}, \mathbf{x}_o) - \mathbb{E}_{\mathbf{x}' \sim p(\mathbf{x}|\boldsymbol{\theta})}[d(\mathbf{x}', \mathbf{x}_o)])(\mathbb{E}_{\mathbf{x}' \sim p(\mathbf{x}|\boldsymbol{\theta})}[d(\mathbf{x}', \mathbf{x}_o)] - g(\boldsymbol{\theta}))].$$
The first term in the products reads a difference of the same term, and, thus, is zero.

Thus, since $X = 0$ and since $\mathbb{E}_{\mathbf{x}' \sim p(\mathbf{x}|\boldsymbol{\theta})}[d(\mathbf{x}', \mathbf{x}_o)] - g(\boldsymbol{\theta})^2 \geq 0$, we have
$$\mathbb{E}_{\theta, \mathbf{x} \sim p(\boldsymbol{\theta}, \mathbf{x})}[(d(\mathbf{x}, \mathbf{x}_o) - g(\boldsymbol{\theta}))^2] \geq \mathbb{E}_{\theta, \mathbf{x} \sim p(\boldsymbol{\theta}, \mathbf{x})}[(d(\mathbf{x}, \mathbf{x}_o) - \mathbb{E}_{\mathbf{x}' \sim p(\mathbf{x}|\boldsymbol{\theta})}[d(\mathbf{x}', \mathbf{x}_o)])^2].$$

$\square$

### A3.2 Proposition 2 and proof

**Proposition 2.** *Let $p(\boldsymbol{\theta}, \mathbf{x})$ be the joint distribution over parameters and data and let $p(\mathbf{x}_t)$ be a distribution of target samples. Let $\ell(\boldsymbol{\theta}; \mathbf{x}_o)$ be a cost function and $f_\phi(\cdot)$ a parameterized function as in proposition 1. Then, the loss function $\mathcal{L} = \mathbb{E}_{p(\boldsymbol{\theta}, \mathbf{x}) p(\mathbf{x}_t)}[(f_\phi(\boldsymbol{\theta}, \mathbf{x}_t) - d(\mathbf{x}, \mathbf{x}_t))^2]$ is minimized if and only if, for all $\boldsymbol{\theta} \in supp(p(\boldsymbol{\theta}))$ and all $\mathbf{x}_t \in supp(p(\mathbf{x}_t))$ we have $f_\phi(\boldsymbol{\theta}, \mathbf{x}_t) = \mathbb{E}_{p(\mathbf{x}|\boldsymbol{\theta})}[d(\mathbf{x}, \mathbf{x}_t)]$.*

*Proof.* We aim to prove that
$$\mathbb{E}_{\theta, \mathbf{x} \sim p(\boldsymbol{\theta}, \mathbf{x}), \mathbf{x}_t \sim p(\mathbf{x}_t)}[(d(\mathbf{x}, \mathbf{x}_t) - g(\boldsymbol{\theta}, \mathbf{x}_t))^2] \geq$$
$$\mathbb{E}_{\theta, \mathbf{x} \sim p(\boldsymbol{\theta}, \mathbf{x}), \mathbf{x}_t \sim p(\mathbf{x}_t)}[(d(\mathbf{x}, \mathbf{x}_t) - \mathbb{E}_{\mathbf{x}' \sim p(\mathbf{x}|\boldsymbol{\theta})}[d(\mathbf{x}', \mathbf{x}_t)])^2]$$
for every function $g(\boldsymbol{\theta}, \mathbf{x}_t)$. We begin as in proposition 1:
$$\mathbb{E}_{\theta, \mathbf{x} \sim p(\boldsymbol{\theta}, \mathbf{x}), \mathbf{x}_t \sim p(\mathbf{x}_t)}[(d(\mathbf{x}, \mathbf{x}_t) - g(\boldsymbol{\theta}, \mathbf{x}_t))^2] =$$
$$\mathbb{E}_{p(\mathbf{x}_t)}[\mathbb{E}_{\theta, \mathbf{x} \sim p(\boldsymbol{\theta}, \mathbf{x})}[(d(\mathbf{x}, \mathbf{x}_t) - g(\boldsymbol{\theta}, \mathbf{x}_t))^2]]$$
Below, we prove that, for *any* $\mathbf{x}_t$, the optimal $g(\boldsymbol{\theta}, \mathbf{x}_t)$ is the conditional expectation $\mathbb{E}_{\mathbf{x}' \sim p(\mathbf{x}|\boldsymbol{\theta})}[d(\mathbf{x}', \mathbf{x}_t)]$:
$$\mathbb{E}_{\theta, \mathbf{x} \sim p(\boldsymbol{\theta}, \mathbf{x})}[(d(\mathbf{x}, \mathbf{x}_t) - g(\boldsymbol{\theta}, \mathbf{x}_t))^2] =$$
$$\mathbb{E}_{\theta, \mathbf{x} \sim p(\boldsymbol{\theta}, \mathbf{x})}[(d(\mathbf{x}, \mathbf{x}_t) - \mathbb{E}_{\mathbf{x}' \sim p(\mathbf{x}|\boldsymbol{\theta})}[d(\mathbf{x}', \mathbf{x}_t)] + \mathbb{E}_{\mathbf{x}' \sim p(\mathbf{x}|\boldsymbol{\theta})}[d(\mathbf{x}', \mathbf{x}_t)] - g(\boldsymbol{\theta}, \mathbf{x}_t))^2] =$$
$$\mathbb{E}_{\theta, \mathbf{x} \sim p(\boldsymbol{\theta}, \mathbf{x})}[(d(\mathbf{x}, \mathbf{x}_t) - \mathbb{E}_{\mathbf{x}' \sim p(\mathbf{x}|\boldsymbol{\theta})}[d(\mathbf{x}', \mathbf{x}_t)])^2 + (\mathbb{E}_{\mathbf{x}' \sim p(\mathbf{x}|\boldsymbol{\theta})}[d(\mathbf{x}', \mathbf{x}_t)] - g(\boldsymbol{\theta}, \mathbf{x}_t))^2] + X$$
with
$$X = \mathbb{E}_{\theta, \mathbf{x} \sim p(\boldsymbol{\theta}, \mathbf{x})}[(d(\mathbf{x}, \mathbf{x}_o) - \mathbb{E}_{\mathbf{x}' \sim p(\mathbf{x}|\boldsymbol{\theta})}[d(\mathbf{x}', \mathbf{x}_o)])(\mathbb{E}_{\mathbf{x}' \sim p(\mathbf{x}|\boldsymbol{\theta})}[d(\mathbf{x}', \mathbf{x}_o)] - g(\boldsymbol{\theta}, \mathbf{x}_t))],$$
which, as above, is $X = 0$ (proof is identical to proposition 1). Thus, since $\mathbb{E}_{\mathbf{x}' \sim p(\mathbf{x}|\boldsymbol{\theta})}[d(\mathbf{x}', \mathbf{x}_t)] - g(\boldsymbol{\theta}, \mathbf{x}_t)^2 \geq 0$, we have:
$$\mathbb{E}_{\theta, \mathbf{x} \sim p(\boldsymbol{\theta}, \mathbf{x})}[(d(\mathbf{x}, \mathbf{x}_t) - g(\boldsymbol{\theta}, \mathbf{x}_t))^2] \geq \mathbb{E}_{\theta, \mathbf{x} \sim p(\boldsymbol{\theta}, \mathbf{x})}[(d(\mathbf{x}, \mathbf{x}_o) - \mathbb{E}_{\mathbf{x}' \sim p(\mathbf{x}|\boldsymbol{\theta})}[d(\mathbf{x}', \mathbf{x}_o)])^2]$$
Because this inequality holds for *any* $\mathbf{x}_t$, the average over $p(\mathbf{x}_t)$ will also be minimized if and only if $g(\boldsymbol{\theta}, \mathbf{x}_t))$ matches the conditional expectation $\mathbb{E}_{\mathbf{x}' \sim p(\mathbf{x}|\boldsymbol{\theta})}[d(\mathbf{x}', \mathbf{x}_o)]$ for any $\mathbf{x}_t$ within the support of $p(\mathbf{x}_t)$.

$\square$

## A4 Further experimental details

### A4.1 Details on training procedure

**Dataset construction**     We generate samples from the prior $\boldsymbol{\theta}_i \sim p(\boldsymbol{\theta})$ and corresponding simulations $\mathbf{x}_i \sim p(\mathbf{x}|\boldsymbol{\theta}_i)$, to obtain parameter-simulation pairs, $\Theta, X = \{\boldsymbol{\theta}_i, \mathbf{x}_i\}_{i=1...N}$. Next, a dataset of target data points, $X_{\text{target}}$, is constructed by concatenating in the batch-dimension: 1) $X$, 2) a random subset of $X$ augmented with Gaussian noise with a specified variance, i.e., $\mathbf{x}_i + \boldsymbol{\epsilon}_i, \boldsymbol{\epsilon}_i \sim \mathcal{N}(0, \sigma^2 \mathbf{I})$, and 3) any number of real experimental observations the user wishes to include, $X_{\text{real}}$; in total, $N_{\text{target}} = N + N_{\text{noised}} + N_{\text{real}}$. Note that $X_{\text{target}}$ does not technically need to include any simulated or real data, only 'realistic' targets in the neighborhood of the observed data that one eventually wants to perform inference for. Practically, 1) has already been simulated, and 2) and 3) does not require additional simulations while improving performance through noise augmentation and having access to real data targets. Finally, a pairwise distance matrix $D$ is computed with elements $d_{i,j} = d(\mathbf{x}_i, \mathbf{x}_t), \mathbf{x}_i \in X, \mathbf{x}_t \in X_{\text{target}}$. $D$ can either be pre-computed in full, or partially within the training loop.

**Network optimization and convergence to GBI loss**     Given dataset $\Theta, X, X_{\text{target}}, D$, a fully connected feed-forward deep neural network with weights $\phi$ is trained to minimize the mean squared error loss: $\frac{1}{N_{\text{sim}} n_{\text{targets}}} \sum_{i=1}^{N_{\text{sim}}} \sum_{t=1}^{n_{\text{target}}} (\text{NN}_\phi(\boldsymbol{\theta}_i, \mathbf{x}_t) - d(\mathbf{x}_i, \mathbf{x}_t))^2$. In other words, for a parameter configuration $\boldsymbol{\theta}_i$ and a target data point $\mathbf{x}_t$, the network is trained to predict the distance $d_{i,t}$ between the corresponding single (stochastic) simulation $\mathbf{x}_i$ and the target $\mathbf{x}_t$. In every training epoch, $n_{\text{target}}$ target simulations (usually 1-10, compared to $N_{\text{target}} > 10000$) are randomly sub-sampled per $\boldsymbol{\theta}_i$, drastically reducing training time and allowing the relevant $d_{i,t}$ to be computed on the fly.

Note that we do not wish to accurately predict the distance $d(\mathbf{x}_i, \mathbf{x}_t)$ in data-space for individual simulations $\mathbf{x}_i$, but rather the loss $\ell(\boldsymbol{\theta}_i; \mathbf{x}_t)$. Conveniently, with mean squared error as the objective function for network training, for a fixed pair of $\boldsymbol{\theta}_i, \mathbf{x}_t$, the trained network predicts the mean distance, $\mathbb{E}_{p(\mathbf{x}|\boldsymbol{\theta}_i)}[d(\mathbf{x}, \mathbf{x}_t)]$, precisely the loss function we target in Eq.2 (proof in Appendix).

### A4.2 Description of benchmark tasks, distance and cost functions

#### A4.2.1 Uniform 1D

A one-dimensional task with uniform noise to illustrate how expected distance from observation can be used as a cost function for inference, especially in the case of model misspecification:

| | |
|---|---|
| **Prior** | $\mathcal{U}(-1.5, 1.5)$ |
| **Simulator** | $\mathbf{x}\|\boldsymbol{\theta} = g(z) + \epsilon$, where $\epsilon \sim \mathcal{U}(-0.25, 0.25), z = 0.8 \times (\boldsymbol{\theta} + 0.25)$, and $x_{\text{noiseless}} = g(z) = 0.1627 + 0.9073z - 1.2197z^2 - 1.4639z^3 + 1.4381z^4$. |
| **Dimensionality** | $\boldsymbol{\theta} \in \mathbb{R}^1, \mathbf{x} \in \mathbb{R}^1$ |
| **Cost function** | $d(\mathbf{x}, \mathbf{x}_o)$: MSE; $p(\mathbf{x}\|\boldsymbol{\theta})$ computed exactly given uniform noise. For obtaining the true cost, Eq. 2 is analytically integrated over $x_{\text{noiseless}} \pm 0.25$ |
| **Posterior** | Ground-truth GBI posterior samples are obtained via rejection sampling. We used the prior as proposal distribution. |

#### A4.2.2 Two Moons

A two-dimensional task with a posterior that exhibits both global (bimodality) and local (crescent shape) structure to illustrate how algorithms deal with multimodality:

| | |
|---|---|
| **Prior** | $\mathcal{U}(-\mathbf{1}, \mathbf{1})$ |
| **Simulator** | $\boldsymbol{x}\|\boldsymbol{\theta} = \begin{bmatrix} r\cos(\alpha) + 0.25 \\ r\sin(\alpha) \end{bmatrix} + \begin{bmatrix} -\|\theta_1 + \theta_2\|/\sqrt{2} \\ (-\theta_1 + \theta_2)/\sqrt{2} \end{bmatrix}$, where $\alpha \sim \mathcal{U}(-\pi/2, \pi/2)$, and $r \sim \mathcal{N}(0.1, 0.01^2)$ |
| **Dimensionality** | $\boldsymbol{\theta} \in \mathbb{R}^2, \mathbf{x} \in \mathbb{R}^2$ |
| **Cost function** | $d(\mathbf{x}, \mathbf{x}_o)$: MSE; and $p(\mathbf{x}\|\boldsymbol{\theta})$ computed exactly. To obtain the true cost, Eq. 2 is numerically integrated over a 2D grid of $d\mathbf{x}$ with 500 equal bins in both dimensions, where $x^{(1)} \in [-1.2, 0.4], x^{(2)} \in [-1.6, 1.6]$. |
| **Posterior** | Ground-truth GBI posterior samples are obtained via rejection sampling with the prior as proposal distribution. |
| **References** | [24, 8] |

### A4.2.3 Linear Gaussian

Inference of the mean of a 10-d Gaussian model, in which the covariance is fixed. The (conjugate) prior is Gaussian:

| | |
|---|---|
| **Prior** | $\mathcal{N}(\mathbf{0}, 0.1 \odot \mathbf{I})$ |
| **Simulator** | $\mathbf{x}\|\boldsymbol{\theta} \sim \mathcal{N}(\mathbf{x}\|\mathbf{m}_{\boldsymbol{\theta}} = \boldsymbol{\theta}, \mathbf{S} = 0.1 \odot \mathbf{I})$ |
| **Dimensionality** | $\boldsymbol{\theta} \in \mathbb{R}^{10}, \mathbf{x} \in \mathbb{R}^{10}$ |
| **Cost function** | $d(\mathbf{x}, \mathbf{x}_o)$: MSE; and $p(\mathbf{x}\|\boldsymbol{\theta})$ computed exactly. The true cost (Eq. 2) can be computed analytically. |
| **Posterior** | Ground-truth GBI posterior samples are obtained following the procedure in Lueckmann et al. [24]: We first ran MCMC to generate 10k samples from the GBI posterior. We then trained a neural spline flow on these 10k samples and, finally, used the trained flow as proposal distribution for rejection sampling. |
| **References** | [24, 8] |

### A4.2.4 Gaussian Mixture

The single-trial version of this task is common in the ABC literature. It consists of inferring the common mean of a mixture of two two-dimensional Gaussian distributions, one with much broader covariance than the other. In this study, we used this task to infer the common mean of the two distributions from five i.i.d. simulations:

| | |
|---|---|
| **Prior** | $\mathcal{U}(-\mathbf{10}, \mathbf{10})$ |
| **Simulator** | $\mathbf{x}\|\boldsymbol{\theta} \sim 0.5\,\mathcal{N}(\mathbf{x}\|\mathbf{m}_{\boldsymbol{\theta}} = \boldsymbol{\theta}, \mathbf{S} = \mathbf{I}) + 0.5\,\mathcal{N}(\mathbf{x}\|\mathbf{m}_{\boldsymbol{\theta}} = \boldsymbol{\theta}, \mathbf{S} = 0.01 \odot \mathbf{I})$ |
| **Dimensionality** | $\boldsymbol{\theta} \in \mathbb{R}^2, \mathbf{x} \in \mathbb{R}^2$ |
| **Cost function** | $d(\mathbf{x}, \mathbf{x}_o)$: MMD$^2$. The true cost (Eq. 2) is obtained by integrating the distance function on a grid for every trial independently and multiplying over the trials. |
| **Posterior** | Ground-truth GBI posterior samples are obtained via rejection sampling. As proposal, we used a Normal distribution centered around the ground truth parameter and with variance $\frac{50}{\beta}$. |
| **References** | [24, 53] |

### A4.3 Training, inference, and evaluation for benchmark tasks

**ACE** For the benchmark tasks, the cost estimation network is a residual network with 3 hidden layers of 64 units [54]. The training dataset was split 90:10 into training and validation sets, with $n_{\text{target}} = 2$ and 5 $\mathbf{x}_t$ randomly sampled each epoch for evaluating training and validation loss, respectively. We used a batchsize of 500, i.e., 500 $\boldsymbol{\theta}$, 2 $\mathbf{x}_t$, for 1000 cost targets per training batch. Networks usually converge within 500 epochs, with 100 epochs of non-decreasing validation loss as the convergence criterion. For inference, we ran multi-chain slice sampling with 100 chains to sample the potential function. For the Gaussian Mixture task with 5 i.d.d. samples per simulation/observation, we appended to the cost estimation network a fully connected, permutation-invariant embedding network [55] (2 layers, 100 units each), which preprocessed the i.d.d. datapoints into a single 20-dimensional vector (but was trained end-to-end).

**Kernel ABC** We compared our algorithm with a version of kernel ABC on the benchmark tasks. For kernel ABC, we accepted samples (from the fixed set of 10000 prior simulations) with probability $\exp(-\beta \cdot d(\mathbf{x}, \mathbf{x}_o))$, where $d(\cdot, \cdot)$ is the distance function. In many cases, and especially for large $\beta$, this yielded very few or no accepted samples. We, therefore, reduced $\beta$ until at least 50 samples were accepted.

**NPE** We used the implementation in the sbi toolbox [56], with neural spline flow [57] as density estimator (five transformation layers) to approximate the posterior. For tasks with bounded priors (all except Linear Gaussian), we appended a sigmoidal transformation to the flow such that its support matches the support of the prior [58]. For the Gaussian Mixture task, the same permutation-invariant embedding net was used as for ACE. Posterior samples were directly obtained from the trained flow. All other hyperparameters were the same as in the sbi toolbox, version 0.19.2 [56].

**NLE and tempered NLE** We used the implementation in the sbi toolbox [56], with neural spline flow [57] as density estimator (five transformation layers) to approximate the likelihood. For the Gaussian Mixture task with 5 i.d.d. simulations per sample/observation, they were split up as 5 independent simulations with the same $\boldsymbol{\theta}$ repeated 5 times, effectively having 5 times the training simulation budget. We ran multi-chain slice sampling with 100 chains to sample the potential function as the sum of log-prior probability and flow-approximated log-likelihood. Tempered NLE uses the same learned likelihood estimator, but the log-likelihood term in the potential function is scaled by $\beta$ during MCMC sampling (same values as used for ACE). All other hyperparameters were the same as in the sbi toolbox, version 0.19.2 [56].

**Noise augmentation** For each task, we randomly subsampled 100 of all simulated datapoints ($\mathbf{x}$), and added Gaussian noise with zero-mean and standard deviation equaling two times the standard deviation of all prior predictives, i.e., $\mathcal{N}(\mathbf{0}, (2\sigma_x)^2\mathbf{I})$. We additionally varied noise amplitude $(0, 2\sigma_x, 5\sigma_x)$ and found little to no effect on benchmark performance (see Appendix A3).

**Synthetic misspecified observations** For the Uniform 1D, 2 Moons, and Linear Gaussian task, we generated an additional 100,000 prior predictive simulations and defined the prior predictive bound as the minimum and maximum (of each dimension) of those simulations and $\sigma_x$ as their standard deviation. Then, to create 20 misspecified observations, we generated 20 model-simulations and added Gaussian noise with zero-mean and standard deviation equaling 0.5 $\sigma_x$ (i.e., $\epsilon \sim \mathcal{N}(\mathbf{0}, (0.5\sigma_x)^2\mathbf{I})$) to the 20 observations iteratively until they are outside of the prior predictive bounds in every dimension. For the Gaussian Mixture task, we replaced the second Gaussian above by $\mathcal{N}(12.5 \times \text{sign}(\boldsymbol{\theta}), 0.5^2\mathbf{I})$, i.e. displacing the mean to the corner of the quadrant whose signs match $\boldsymbol{\theta}$, and slightly increasing the variance.

**Metrics** For the metrics in Fig. 3, in columns 1, 2, and 4 and all corresponding figures in the Appendix, results were aggregated across the 10 well-specified and misspecified samples separately, where marker and error bars represent mean and standard deviation over 10 observations. Columns 1 and 2 show average distance between observation and 5000 posterior predictive simulations from each algorithm (except ABC, for which there were 50 posterior samples). Column 3 shows, for each of the 10 well-specified observations and 3 $\beta$ values, 3 random posterior samples for which we compare the ACE-estimated and ground-truth cost, i.e., $10 \times 3 \times 3 = 90$ points are shown for each

task. Column 4 shows C2ST score between 5000 ground-truth GBI posterior samples and ACE GBI posterior samples.

### A4.4    Training procedure for Hodgkin-Huxley model

For NPE, we used the implementation in the sbi toolbox [56]. As density estimator, we used a neural spline flow [57] with five transformation layers. We used a batchsize of 5,000. We appended a sigmoidal transformation to the flow such that its support matches the support of the prior [58]. All other hyperparameters were the same as in the sbi toolbox, version 0.19.2 [56].

For ACE, we used a residual neural network [54] with 7 layers and 100 hidden units each. We used 10% of all simulations as held-out validation set and stopped training when the validation set did not decrease for 100 epochs. We used a batchsize of 5000. For sampling from the generalized posterior, we ran multi-chain slice sampling with 100 chains. As distance function, we used mean-absolute-error, where the distance in each summary statistic was z-scored with the standard deviation of the prior predictives. To generate $\mathbf{x}_t$, we used all simulations which generated between 5 and 40 spikes (since any reasonable experimental recording would fall within this range) and we appended 1000 augmented simulations to which we added Gaussian noise with two times the standard deviation of prior predictives that produce between 5 and 40 spikes. We did not use the experimental recordings from the Allen Cell Types database in $\mathbf{x}_t$.

## A5    Hodgkin-Huxley model

We used the same model as Gonçalves et al. [10], which follows the model proposed in Pospischil et al. [34]. Briefly, the model contains four types of conductances (sodium, delayed-rectifier potassium, slow voltage-dependent potassium, and leak) and has a total of eight parameters that generate a time series which we reduce to seven summary statistics (spike count, mean resting potential, standard deviation of the resting potential, and the first four voltage moments mean, standard deviation, skew, curtosis, same as in Gonçalves et al. [10]).

## A6    Supplementary figures

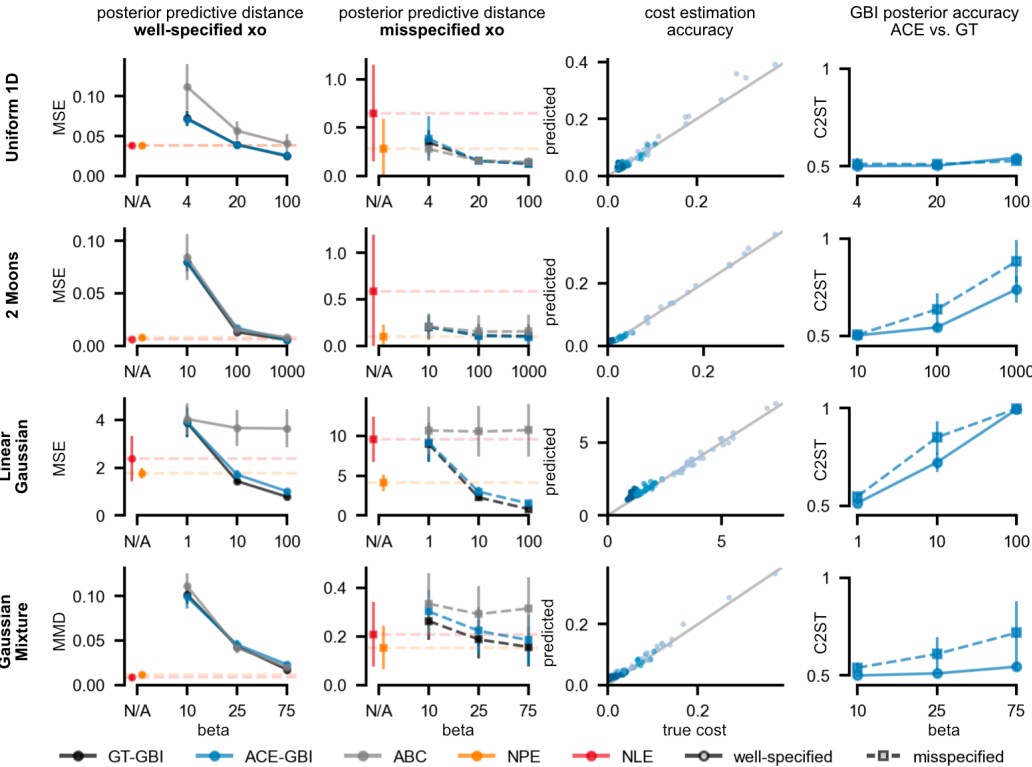

Figure A1:  **Benchmark performance for 'seen' observations** Panels are the same as in Fig. 3, but for the 20 $x_o$ that were included in the target dataset during training.

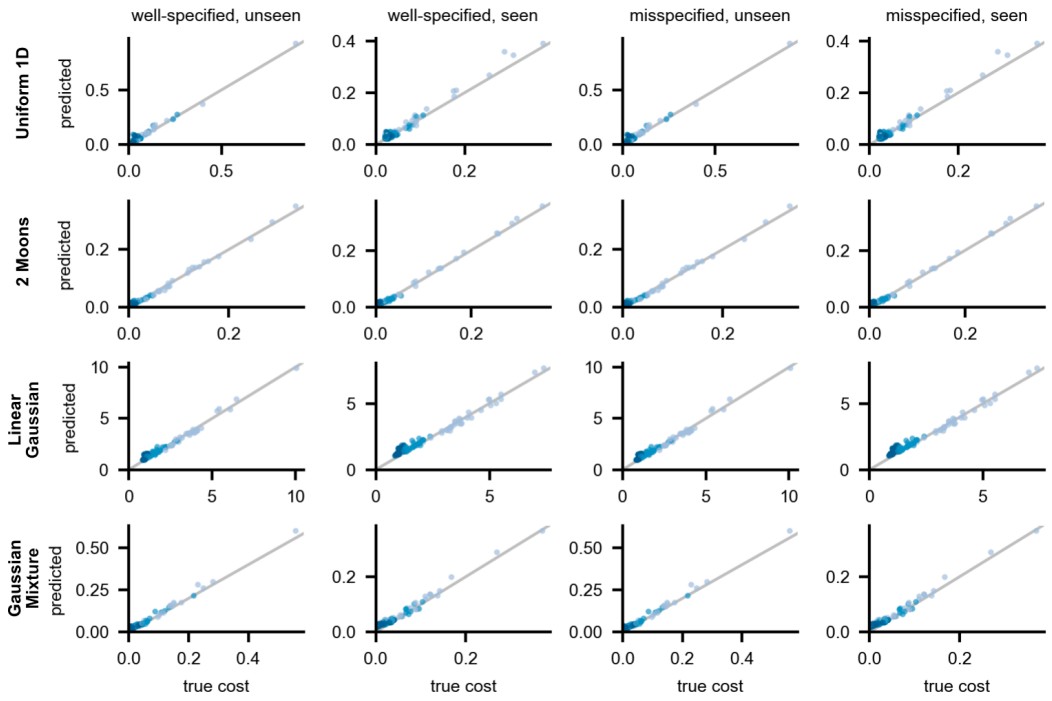

Figure A2: **Cost estimation accuracy for all observations** Columns are same as 3rd column in Fig. 3, but for all combinations of (unseen, seen) and (well-specified, misspecified) observations (10 each, 40 total) for each task. 1st column here is identical to Fig. 3 3rd column.

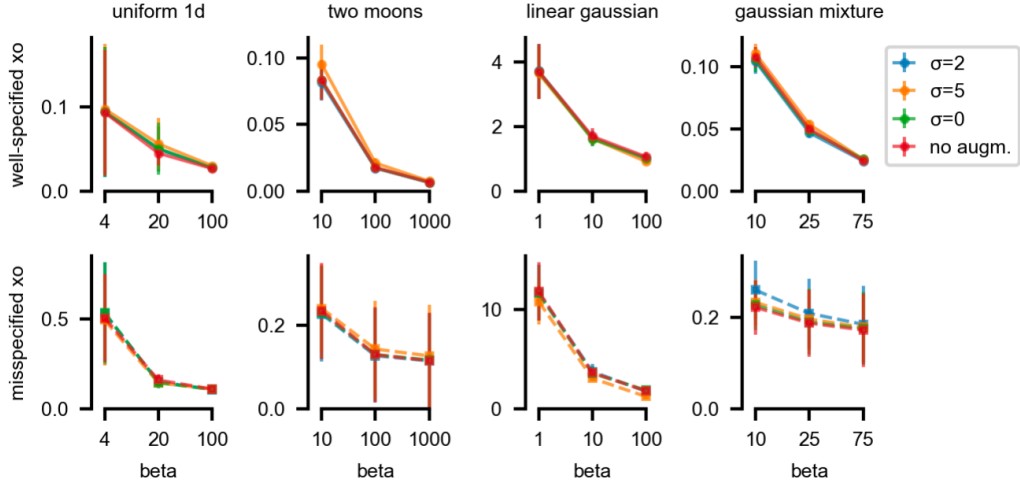

Figure A3: **Benchmark performance for ACE trained with various augmentation noise levels.** Results based on 10k training budget plus 100 noise augmented samples with varying noise amplitudes (blue, orange, green), as well as removing data augmentation altogether (i.e., no noise augmented simulated nor observed data, red). Overall, changing $\sigma$ does not significantly impact performance compared to the original results ($2\sigma_x$, blue).

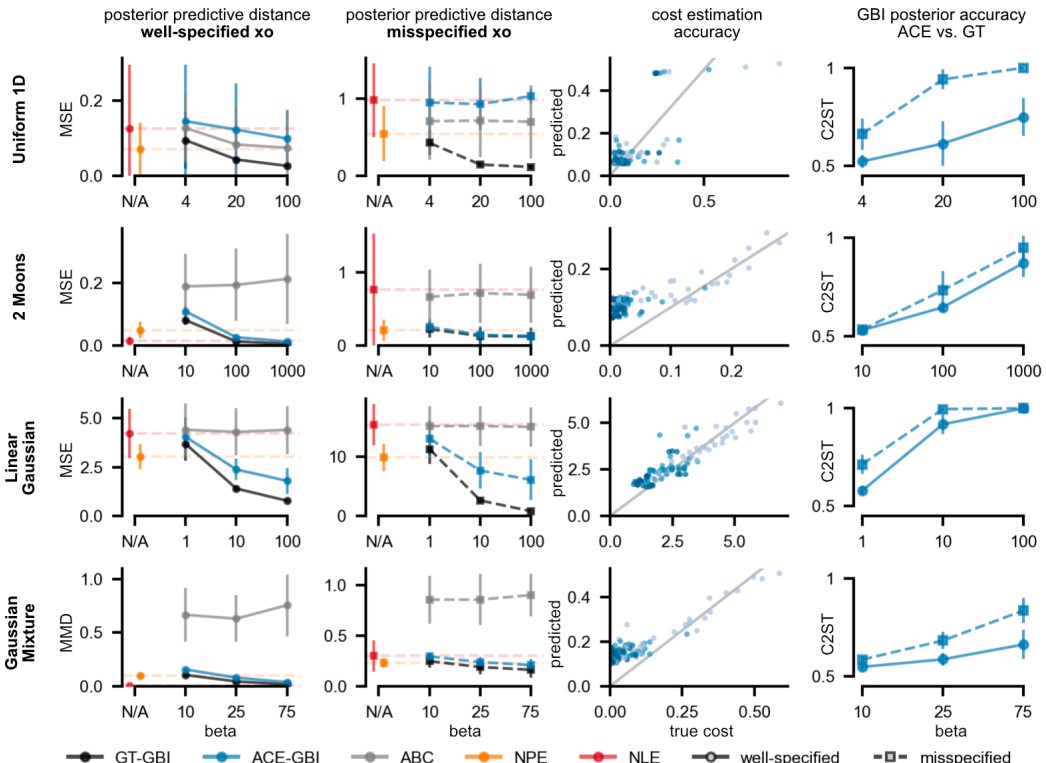

Figure A4: **Benchmark performance with simulation budget of 200** Panels are the same as in Fig. 3, but all algorithms trained with simulation budget of 200.

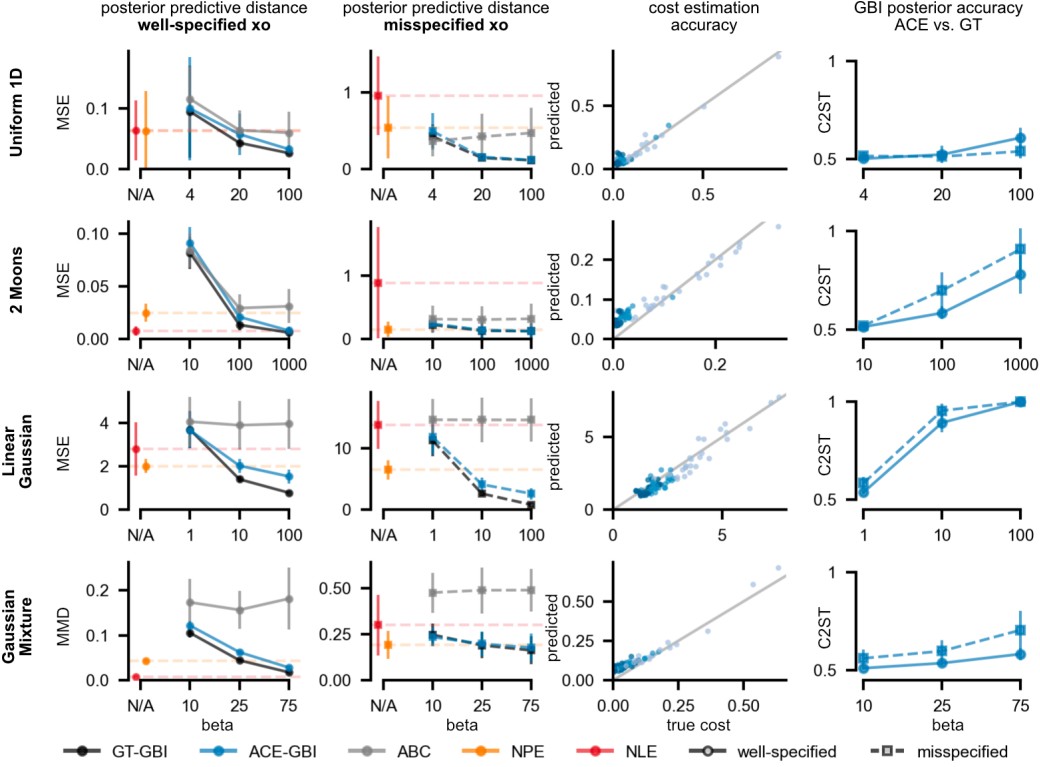

Figure A5: **Benchmark performance with simulation budget of 1000** Panels are the same as in Fig. 3, but all algorithms trained with simulation budget of 1000.

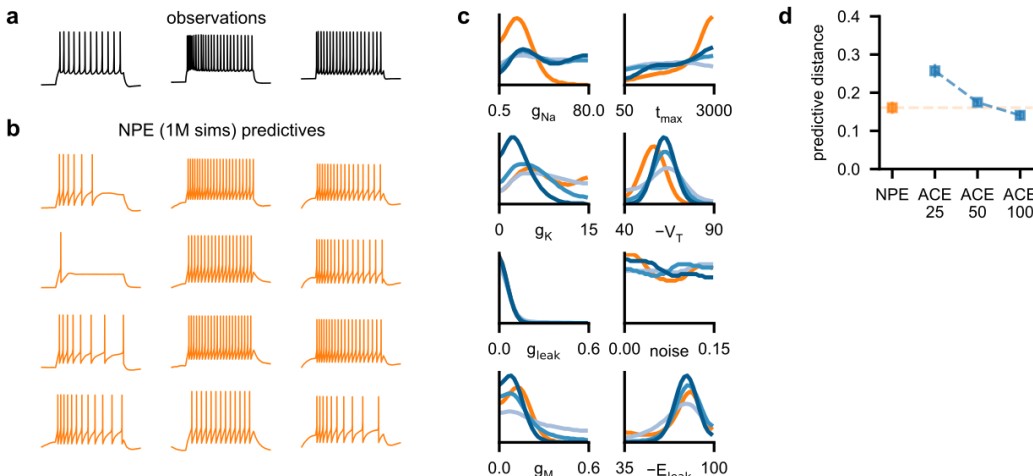

Figure A6:  **NPE performance with 1 million simulations.** Panels are the same as in Fig. 4, but NPE was run with 1 million simulations. ACE is still run with 100K simulations and, thus, the ACE data is the same as in Fig. 4.

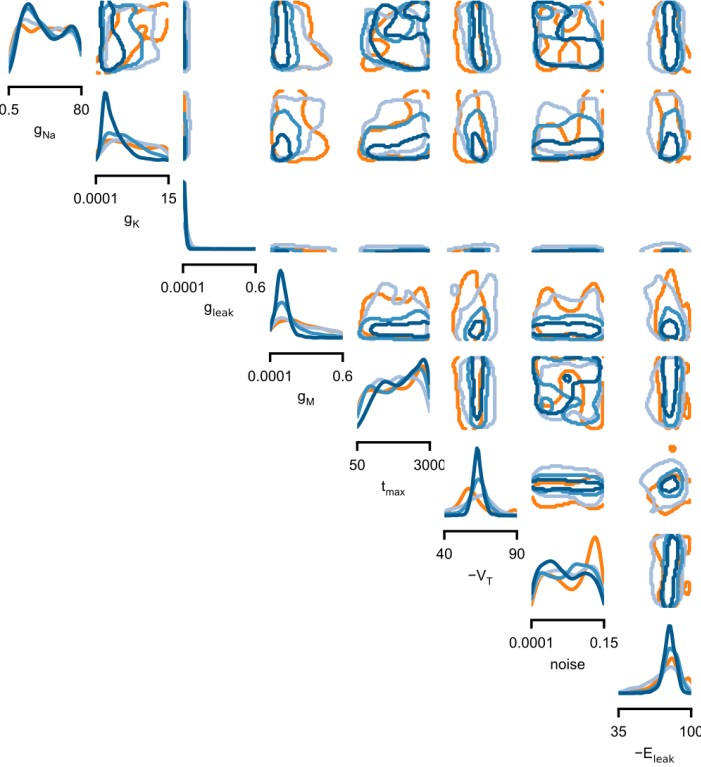

Figure A7:  **Cornerplot of posterior distributions.** Diagonals are 1D-marginals, upper diagonals are 2D-marginals (68th-percentile contour from 5000 posterior samples). Orange: NPE with 100K simulations. Shades of blue: ACE with 100K simulations with $\beta = 25$ (light blue), $\beta = 50$ (medium blue), $\beta = 100$ (dark blue).

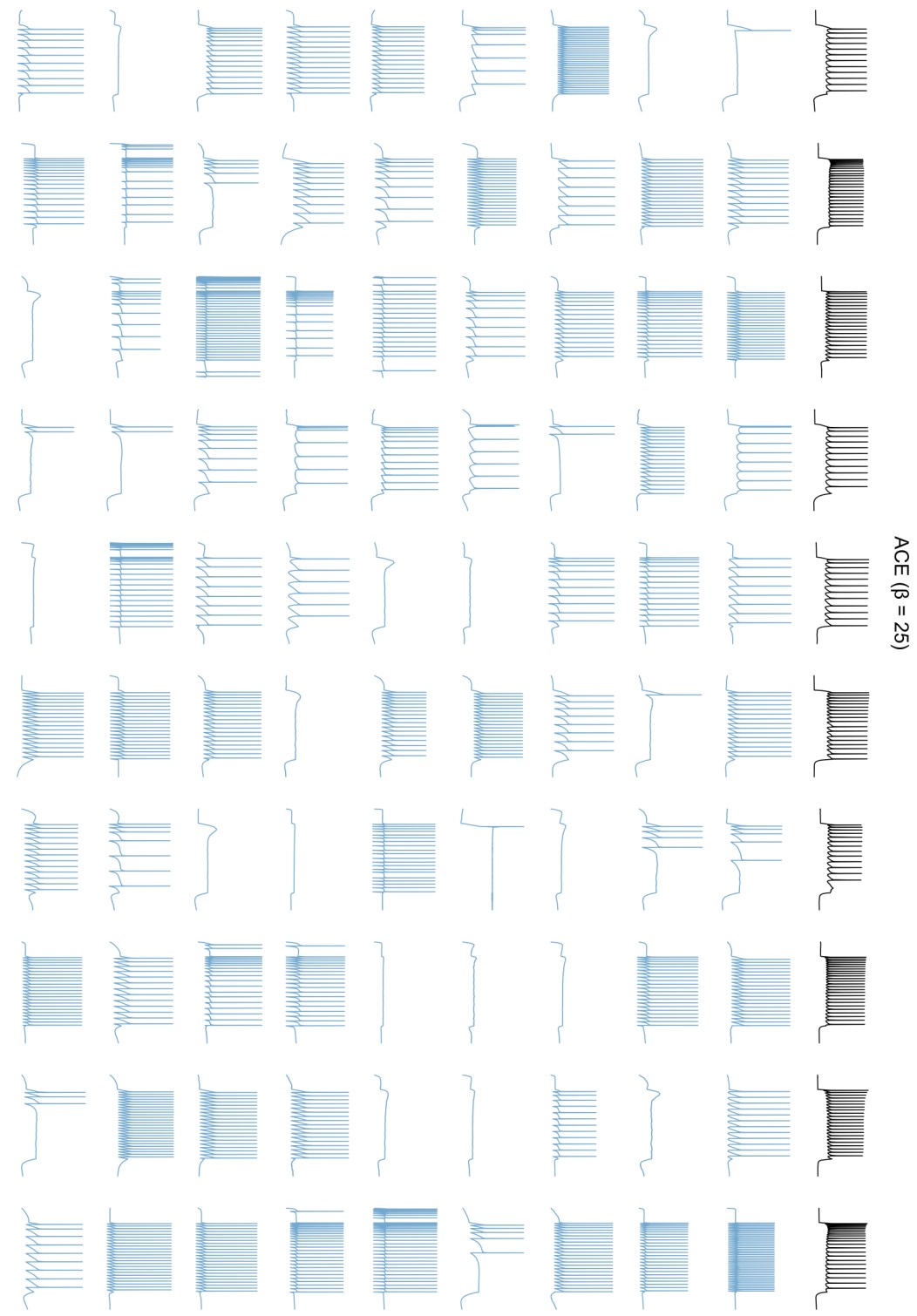

ACE (β = 25)

Figure A8: **Posterior predictive samples of ACE (with 100K simulations) with** $\beta = 25$**.** Top row (black): 10 experimental recordings from the Allen Cell Types database. Below: nine predictive samples given each of the ten observations.

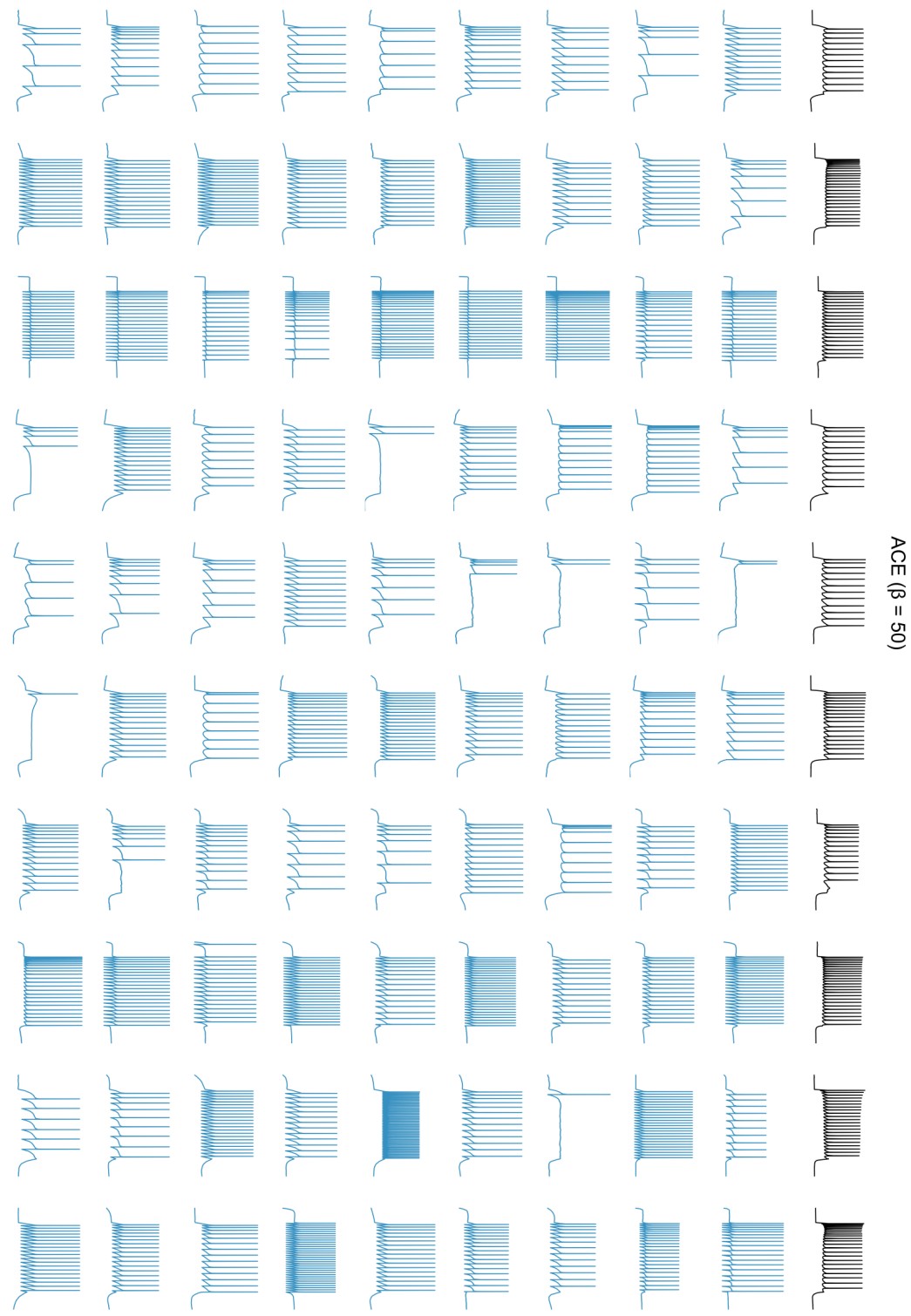

ACE (β = 50)

Figure A9: **Posterior predictive samples of ACE (with 100K simulations) with** $\beta = 50$**.** Top row (black): 10 experimental recordings from the Allen Cell Types database. Below: Nine predictive samples given each of the ten observations.

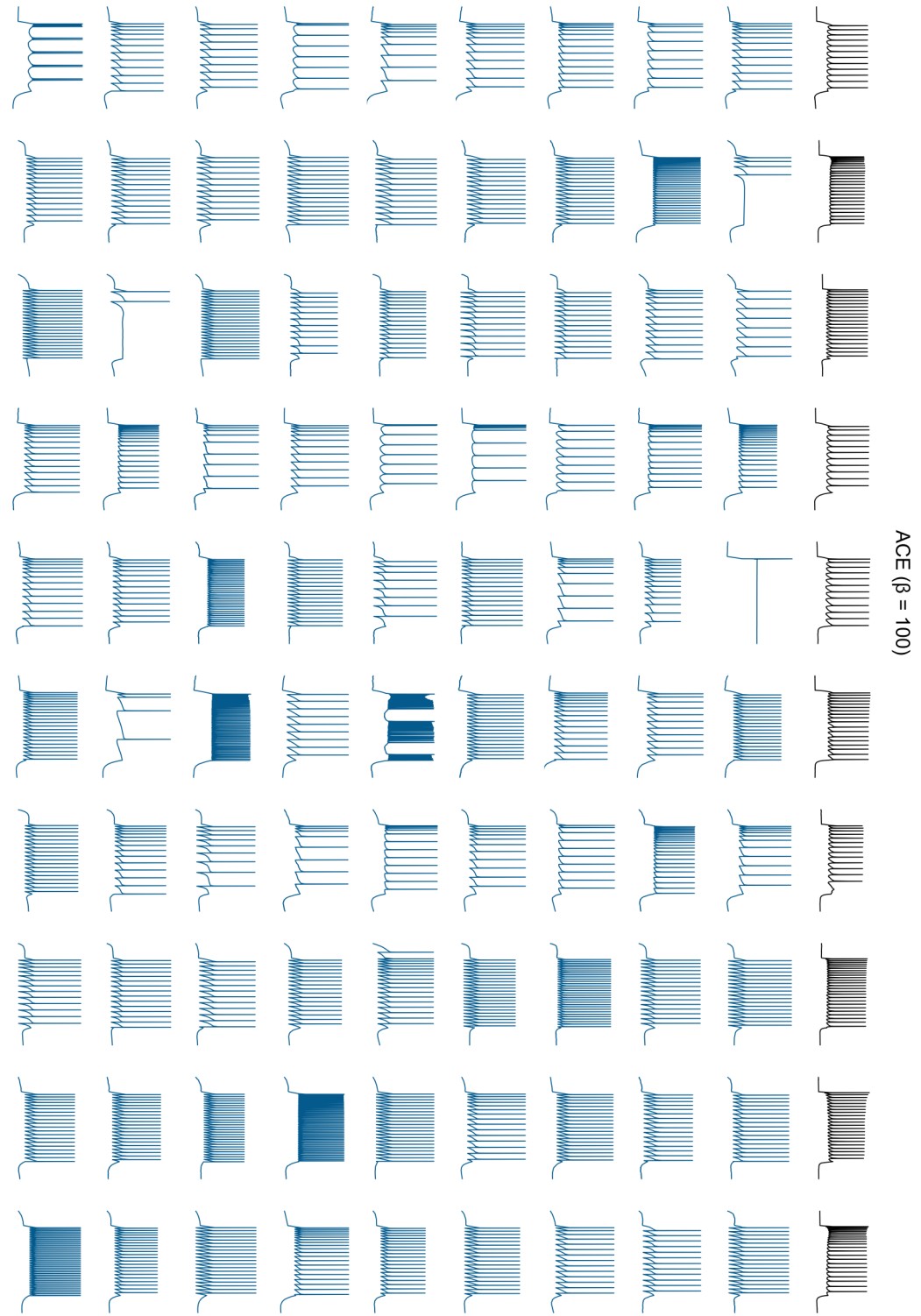

ACE (β = 100)

Figure A10: **Posterior predictive samples of ACE (with 100K simulations) with** $\beta = 100$**.** Top row (black): 10 experimental recordings from the Allen Cell Types database. Below: Nine predictive samples given each of the ten observations.

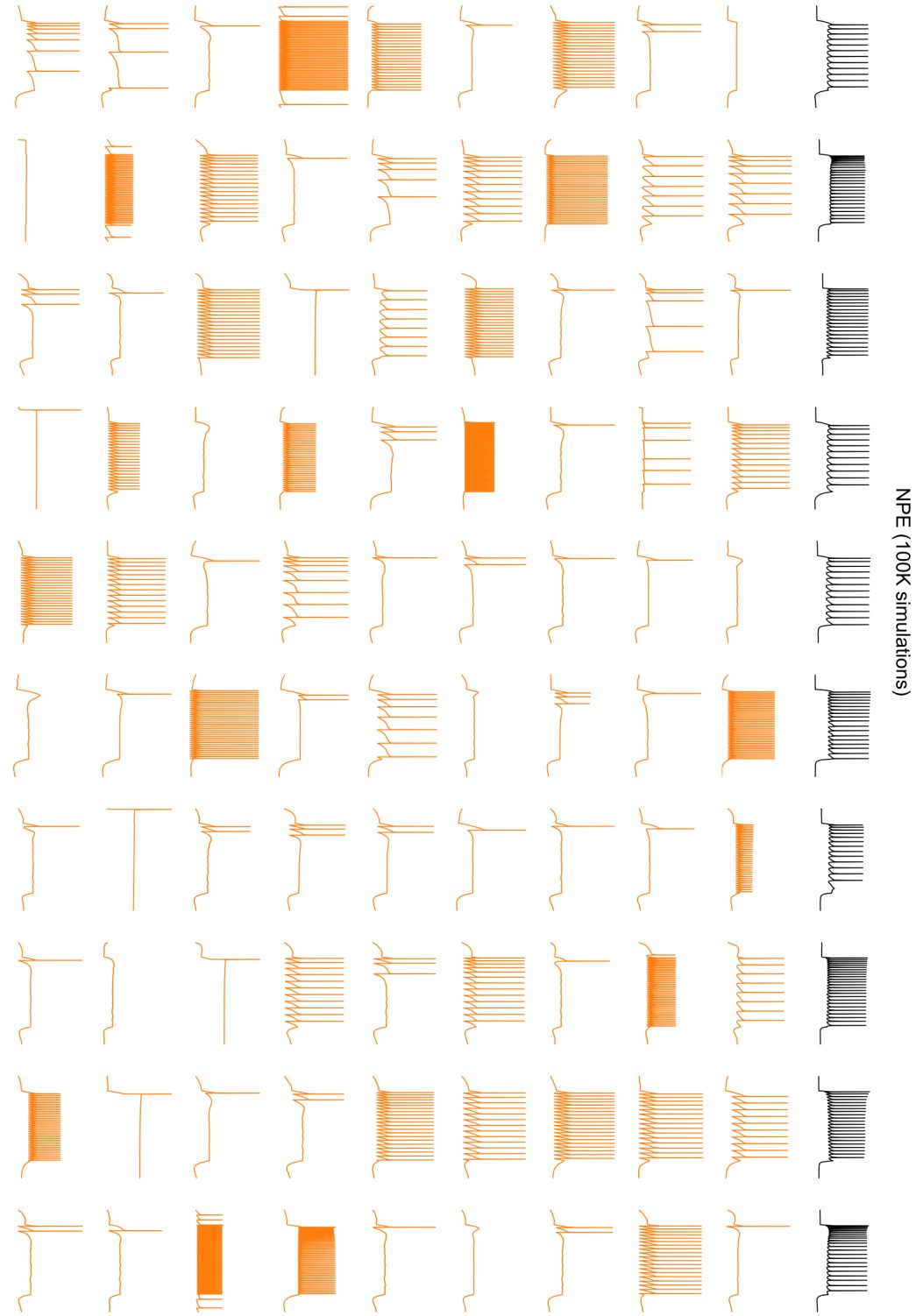

NPE (100K simulations)

Figure A11: **Posterior predictive samples of NPE with 100K simulations.** Top row (black): 10 experimental recordings from the Allen Cell Types database. Below: Nine predictive samples given each of the ten observations.

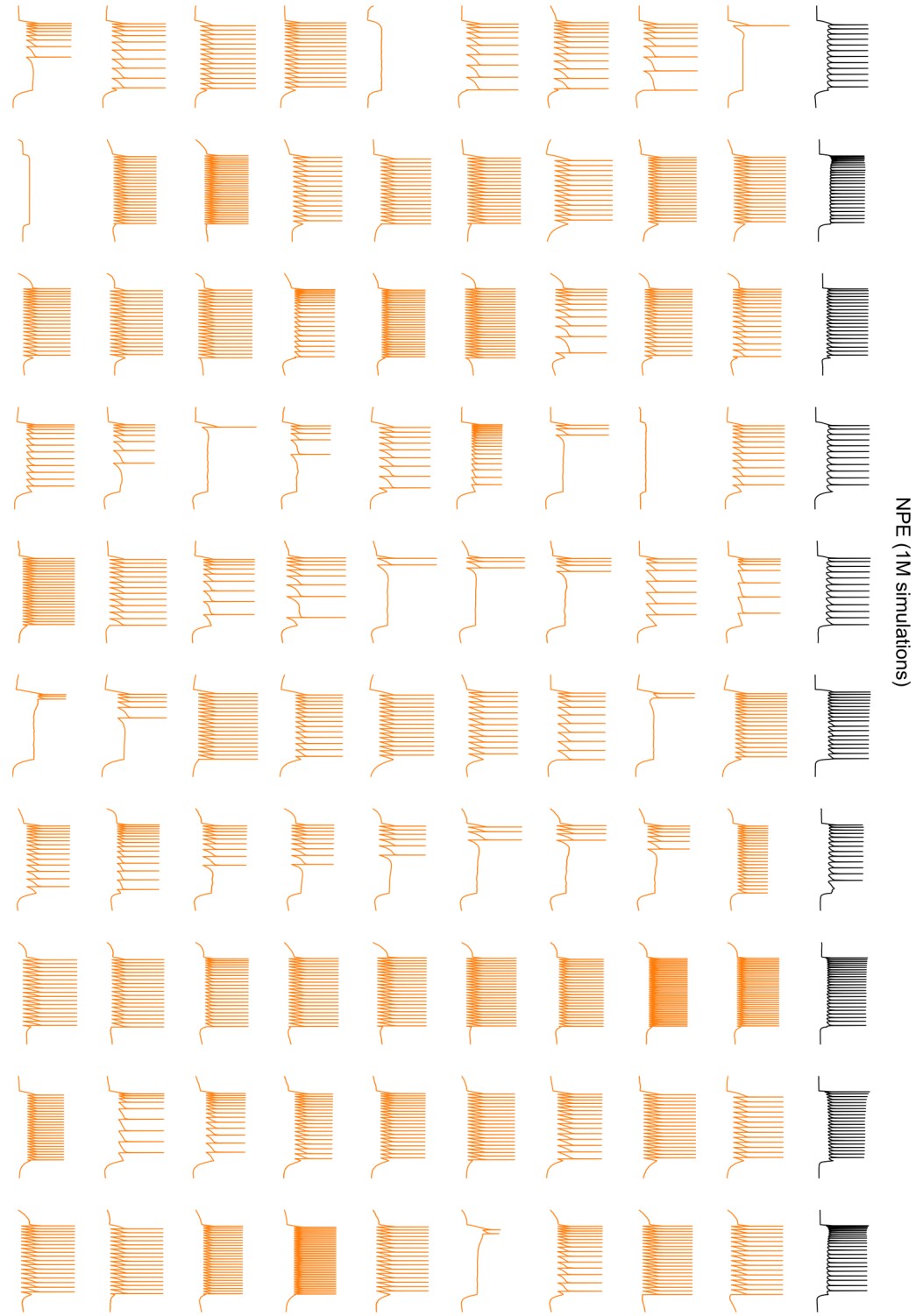

Figure A12: **Posterior predictive samples of NPE with 1M simulations.** Top row (black): 10 experimental recordings from the Allen Cell Types database. Below: Nine predictive samples given each of the ten observations.

