# OpenReview forum: "Generalized Bayesian Inference for Scientific Simulators via Amortized Cost Estimation"
_NeurIPS.cc/2023/Conference — NeurIPS 2023 poster_

### Official Review · Reviewer_9qMX · 2023-06-27

**Soundness:** 3 good
**Presentation:** 4 excellent
**Contribution:** 3 good
**Rating:** 7
**Confidence:** 3

**Summary:**

The authors present a novel approach for SBI by utilizing neural networks (NN) to estimate a generalized cost in GBI. Their proposed method, ACE, demonstrates superior computational efficiency compared to previous approaches, without compromising competitive performance across various evaluation metrics. The authors conducted extensive benchmarking of their method under diverse experimental settings, ranging from 1D to 10D. Furthermore, they conducted experiments using real intracellular recordings, adding an additional dimension to their research.

**Strengths:**

The paper is well written and maintains a smooth flow throughout. Furthermore, the contribution made by the authors appears to be a novel and valuable improvement upon existing ideas in the field. Moreover, the authors have thoughtfully acknowledged and addressed the limitations of their work, placing their findings within the broader context of the field. It is worth noting that their code repository is well-organized, adding to the overall credibility and reproducibility of their research.

**Weaknesses:**

1. To enhance clarity, it would be beneficial to improve the visual presentation of Figure 2. When referring to subfigure C, the arrangement could be adjusted to prevent the automatic focus on the third pane.
2. Propositions 1 and 2 appear somewhat forced, as their results do not seem entirely compelling or noteworthy. It may be worth revisiting these propositions. In the appendix the authors state, "The proofs closely follow standard proofs that regression converges to the conditional expectation." Thus highlighting the lack of importance of these propositions.
3. A minor typo on line 116, "parameter- or".

**Questions:**

nothing, great work!

**Limitations:**

The authors do address the limitations of their work.

---

> ### Author Rebuttal · Authors · 2023-08-08
>
> We really appreciate the reviewer’s encouraging and positive comments about our work, in particular highlighting the novelty and value of our proposed contribution to SBI, our extensive benchmark experiments, and the quality of the writing and code for better communication and reproducibility.
>
> In response to their suggestions regarding clarity, we will:
>
> - Rearrange Figure 3C such that the first panel is bigger and more prominent to prevent automatic focus on the third panel,
> - Move the propositions to the Appendix, such that in the main text we keep the presentation more concise and just state the well-known convergence proof, and refer to the Appendix for more details.
> - Fix the noted typo (thanks for the detailed read!)

---

> > ### Comment · Reviewer_9qMX · 2023-08-14
> >
> > Thank you for the comments.

---

### Official Review · Reviewer_HKbp · 2023-07-05

**Soundness:** 3 good
**Presentation:** 3 good
**Contribution:** 2 fair
**Rating:** 6
**Confidence:** 3

**Summary:**

This paper studies the problem of simulation-based inference - which can is encountered in a wide range of scientific problems - where one is interested in performing Bayesian inference using simulators with implicit likelihoods. Scientific problems can have two unique properties - a) the predictive quality of the simulation is more important than the posterior b) the model can be misspecified. These can be handled though Generalized Bayesian Inference - which generalizes traditional Bayesian Inference by admitting generalized likelihood functions which are defined by some general loss / cost function defined on the parameters. A challenge with employing Generalized Bayesian Inference in the simulation-based case is that estimating the cost function can be computationally expensive. The authors propose learning an amortized neural network estimator for the cost function to reduce the overhead for enabling GBI in simulation-based inference. This amortized cost estimator can be combined with standard inference algorithms (MCMC) to sample from the Generalized Bayesian posterior. The overall proposed approach consists of 3 stages: 1) Collecting data 2) Training ACE 3) Sampling using ACE. The authors present several experiments on a variety of benchmark simulation-based inference tasks and present a case study with the Hodgkin-Huxley simulator.

**Strengths:**

* The problem of simulation-based inference is an important one for various scientific problems and misspecification is an often under-studied but important aspect in practice. The paper attempts to address this important issue relevant to various communities.
* The proposed approach is novel within the context of the simulation based inference. (Although as I mention below, similar ideas have been employed is some related areas)
* Even despite some caveats that I discuss below, the idea is quite neat, and relatively simple (which is a good thing!) Replacing the estimation of the cost with an amortized predictor seems like a natural extension to enable generalized posteriors in the simulation-based inference setting. The simpler algorithm also makes it easier to adopt by practitioners with little overhead.
* The experiments cover a fairly wide variety of tasks used in prior work in the SBI setting. The case study on the HH model was a nice example of leveraging the method on realistic tasks.
* The presentation in the paper was generally quite clear. The paper is well written and easy to follow.
* I also appreciate the authors open-sourcing the code to aid reproducibility.

**Weaknesses:**

* A fundamental weakness I see with the method is that it relies on the neural estimator to generalize well from finitely many simulations. This might not be true in practice (certainly isn’t guaranteed). Put differently, an implicit assumption for the approach to work is that the ACE is trained with _enough_ data to be able to provide informative guidance to the sampler learned in the next stages. As with other work relying on NN estimators this is a fundamental failure mode.
* On a similar note, the method appears to be practical only for relatively low-dimensional problems. The NN learning issues would crop up if the dimensionality of the problem is increased.
* Similar ideas of learning amortized neural estimators have been recently explored in the uncertainty estimation literature [1,2]. In particular, the secondary predictor learned in [1] amortizes a similar quantity.

[1] DEUP: Direct Epistemic Uncertainty Prediction. Lahlou et al., TMLR 2023.

[2] Epistemic Neural Networks. Osband et al., 2023.

**Questions:**

* There are already some results with different simulation budgets - but a more careful study of the effect of the number of samples available to train the ACE might be quite useful.
* Could you comment on the connections to the work on uncertainty estimation mentioned above?
* Do the authors have any thoughts on extending the approach to higher dimensional settings?

**Limitations:**

The authors already discuss several limitations in the paper including a) need for another method to sample from the posterior induced by ACE, b) accommodation of arbitrary distances, c) additional hyper parameters.

In addition to these I would also add access to sufficient samples for training ACE and being limited to low dimensional settings as limitations.

---

> ### Author Rebuttal · Authors · 2023-08-08
>
> We thank the reviewer for their positive assessment of our work, who was very kind to note the importance, novelty, and simplicity of the contribution, as well as our effort on presenting the idea clearly and with included code for reproducibility. We enjoyed the concise and accurate summary of the problem setting surrounding scientific simulators, and we agree that simpler algorithms (with good performance) are more likely to be adopted by practitioners.
>
> We address the reviewer’s questions and suggestions point by point here:
>
> **Problem of neural network estimation with finite training data**:
> We agree wholeheartedly with the reviewer’s concern that the reliance of neural networks on having enough training data is a fundamental limitation of our (and all other) neural network-based SBI algorithms. We will explicitly acknowledge this as a limitation in the discussion. We also agree that a careful characterization of performance vs. simulation budget is a critical sensitivity to measure for any algorithm. We demonstrate precisely the reviewer’s point in our experiments: with a simulation budget of 200 (Suppl. Fig. A3), ACE’s cost estimation accuracy is poor, even though posterior predictive distance is still on par or better compared to other neural methods in this low-training data regime. With a moderate increase in simulation budget (to 1000, Suppl. Fig. A4), we already see a marked improvement in cost estimation. This provides guidance on the simulation budget for problems with similar parameter and data dimensionality. We additionally characterize this dependence in our real-world problem, noting that ACE achieves very good performance compared to NPE even with 10 times less training data (100k vs. 1 million), and is further improved with 1 million training samples.
>
> **Extending to higher-dimensional problems**:
> We acknowledge that here we only provide evidence for good performance in low to moderate dimensionality problems (up to 10 parameters), and we agree that the finite data issue will be further exacerbated in high-dimensional problems. We will include these points in the discussion of limitations as well. However, we note that thus far most successful applications of simulation-based inference are in regimes of parameter dimensionality around 10. For example, the benchmark tasks presented in Lueckmann et al. 2019 all have parameter dimensionality smaller or equal to 10.
>
> In addition, compared to methods such as NLE and NPE, ACE casts the density estimation problem into a regression problem, where the output of the neural network is always kept as 1-dimensional. Therefore, on a task with high dimensionality, e.g., 30 parameters and 30 data dimensions, NPE needs to learn a rather complex 30-D to 30-D transformation, which can be challenging for a normalizing flow with invertibility constraints, whereas ACE only performs a 60-D to 1-D regression, drastically simplifying the problem. Other approaches employed by existing methods, such as adding a preprocessing/embedding network to first reduce parameter and/or data dimensionality, or extending the algorithm to perform sequential inference that concentrates on a specific observation, can be applied here as well. We will include the above in the discussion section, and we thank the reviewer for raising the possibility of future works along these lines.
>
>
> **Relations to work on uncertainty estimation**:
> Thanks for bringing to our attention a related area of literature. From our understanding (and correct us if we’re wrong), both referenced papers train neural networks to predict the _predictive_ uncertainty of a neural network. The “standard” deep NN solves a regression task by predicting the expected (mean) output given an input, while the uncertainty networks in the cited works additionally predict deviations from that, noting various sources of uncertainty (e.g., approximation uncertainty vs. misspecification/epistemic uncertainty). As noted in Lahlou et al 2023, this encapsulates Bayesian NNs, which produce samples from the posterior predictive distribution directly. Furthermore, since the predictive uncertainty can be computed per input without retraining, it is amortized.
>
> Conditional density estimation indeed tackles a similar problem, i.e., predicting not just the mean output, but (samples from) a conditional distribution. Furthermore, standard methods in SBI (such as NLE) currently do not naturally account for misspecification. Therefore, both uncertainty-aware / epistemic neural networks and ACE-GBI are relevant for the high-level problem of producing a diversity of good predictive samples under possible misspecification.
>
> On the other hand, they have very different motivations and regimes of operation, i.e., inverting (misspecified) stochastic mechanistic simulator models vs. giving black-box neural networks awareness of sources of uncertainty. As such, our overall goal, the implementation of amortized estimation of the predictive distance, and its usage in SBI are all quite different from the referenced works on amortized prediction of uncertainty in deep NNs (predictive uncertainty vs. parameter uncertainty, targeting cost vs. posterior variance, using MCMC for sampling vs. directly predicting the full distribution).
>
> Nevertheless, there are interesting potential connections between predictive uncertainty, Bayesian NNs, and GBI, which can be explored in future works for robustifying both deep NN and SBI approaches to misspecification and other sources of uncertainty. We thank the reviewer for raising these potential connections, and will note the above discussion in the paper as well.

---

> > ### Comment · Reviewer_HKbp · 2023-08-15
> > **Response to author rebuttal**
> >
> > Thank you for the response and sorry for the late response.
> >
> > > Relations to work on uncertainty estimation
> >
> > Thanks for the elaborate explanation. To clarify, my comment on the uncertainty estimation methods was merely to highlight the connection between the two, not necessarily as a shortcoming. Apologies for the miscommunication. I think a concise version of this summary would be good to have in the paper.
> >
> > I appreciate the authors responses which partially address the concerns I raised about the reliability of NNs trained on finite data and high-dimensional problems. But these fundamental limitations still remain so I will keep my score, recommending acceptance.

---

### Official Review · Reviewer_CFba · 2023-07-05

**Soundness:** 3 good
**Presentation:** 3 good
**Contribution:** 3 good
**Rating:** 6
**Confidence:** 3

**Summary:**

This paper proposes amortized cost estimation (ACE) for generalized Bayesian inference (GBI) for SBI. The paper trains a neural network to approximate the cost function. The paper demonstrates results on baseline synthetic SBI examples followed by a real-world application using experimental data from the Allen Cell Types Database.

**Strengths:**

* The paper is written clearly and is structured well.
* The idea appears novel. While, GBI was previously defined in [13], its application within the SBI literature is interesting and is therefore of value to the community.
* Algorithm box makes the implementation of the algorithm clear to the reader.


**Weaknesses:**

* The main weakness of the paper seems to be the experimentation. While the figures are neatly presented, it is not that clear what the reader is supposed to conclude from the results:
    * For example, it was not clearly defined what the difference between specified and misspecified samples means within Figure 3. It would be helpful to mathematically define these differences rather than describe them within the text.
    * What is the definition of the GT-GBI? It first appears on line 173 and it is not clear what the true distance is and how it is obtained. This misunderstanding makes it further difficult to understand Figure 3.
    * Another question is why are the first three rows only showing MSE and the last row showing MMD? This is confusing and makes one question why the Figure is not consistent with the metric.
    * Additionally, for the MSE/MMD it seems that as $\beta$ is increased, the MSE/MMD goes down, but the C2ST seems to go up. Why is this the case? What happens in the limit that $\beta$ goes to infinity?


**Questions:**

* Compared to BGI, equation (2) seems to not include the KL divergence term. Can this approach really be called GBI if it is not being regularized by some prior? Why was the KL term not included?
* Following on from the question regarding the prior, is the addition of the noise ($\epsilon$) implicitly adding a prior? It might be the case that adding this noise is also enforcing some implicit distance metric between nearby parameters. I.e. does it assume that there is a local Euclidean distance metric?
* Why does this approach require multi-chain slice-sampling? Is random-walk Metropolis-Hastings not sufficient?


**Limitations:**

* The idea behind the paper is interesting, but the experimental results are difficult to interpret. Further clarifications on the above would help increase the score.

---

> ### Author Rebuttal · Authors · 2023-08-08
>
> We thank the reviewer for noting the novelty of our work and the clear structure of the paper, and their positive score of 3s (good) in soundness, presentation, and contribution. We appreciate their requests for further clarifications and the opportunity to increase the score. We apologize for the confusion caused by the lack of explicitly presented information in these cases. We address each concern below and will modify the paper accordingly. We hope this clarifies the reviewer’s concerns and that it allows them to recommend our paper for acceptance.
>
> **Difference between well-specified and misspecified observations**:
> This was a clear oversight on our part, as we failed to refer to Appendix A4.3 on line 615 in the main text, which contains the mathematical definitions for the synthetic well-specified and misspecified observations. We do so now in the main text and expand on the original description, and summarize A4.3 here for convenience:
>
> Well-specified observations for all tasks were prior simulations. For the Uniform 1D, 2 Moons, and Linear Gaussian task, misspecified observations were prior simulations that were successively perturbed with Gaussian noise with fixed variance (i.e., Gaussian random walk) until the sample was outside the range (i.e., [min, max]) defined by 100k prior simulations in all dimensions. For the Gaussian Mixture task, misspecified samples were generated by replacing the second of the two Gaussians in the simulator with N(12.5 * sign(theta), 0.5^2*eye).
>
> **Definition of GT-GBI and true distance**:
> We apologize for, and will correct the mistake on line 172: It should have been “true cost”, not “true distance”. Throughout the paper, we use “distance” to refer to the output of the distance function (e.g., MSE) between two points in data space, while “cost” refers to the _expectation of the distance_ between all simulations generated by a parameter (theta) and an observation in data space (x). We compute the ground-truth cost function (Eq. 2) for all benchmark tasks, either computing the integral analytically, or numerically via quadrature (i.e., summing over a fine grid). This is then used to define the unnormalized GBI posterior in Eq. 1 and sampled using MCMC or rejection sampling, referred to as GT-GBI samples. This will be clarified in the main text.
>
> **MSE vs. MMD in Fig. 3**:
> 3 of the benchmark tasks use MSE as the distance function in data space, since Euclidean distance between two points are easily measured. In the fourth task (Gaussian Mixture), each observation is a set of 5 independently sampled data points, therefore the distance function must measure the statistical distance between two distributions (which MSE cannot, but MMD can). In addition, users may want to use different distance functions, and it is a feature of ACE that it works for a diverse set of distance functions. The first and second columns of Fig. 3 quantify the average distance the posterior predictive simulations achieve for each of the algorithms, hence the first 3 rows have the y-axis labeled as MSE, and the last as MMD. More details in Appendix A4.2.
>
> **MSE/MMD goes down while C2ST goes up with increasing beta**:
> Due to the exponential in Eq. 1, as beta increases, the GBI posterior becomes more concentrated near parameter regions with low cost. Therefore, high-beta posterior predictive simulations have lower distance to the observation (thus lower MSE & MMD). At the same time, since the cost estimation network only learns to approximate cost using finite simulations, if the ground truth posterior is very narrow, then small errors in cost estimation can lead to large changes in C2ST, resulting in generally higher C2ST with increasing beta.
>
> In the limit of beta approaching infinity, the true GBI posterior would collapse onto the minimizer of the cost function (i.e., the parameter that produces simulations with the lowest average distance to the observation). Samples from this ground-truth posterior would achieve the lowest cost by definition, while samples from the ACE posterior would be close, but slightly higher in cost, and the two sets of samples coming from two different delta functions would be perfectly separable, resulting in a C2ST score of 1 but comparably small predictive distances. We will expand on the current discussion of this in the results section (line 235).
>
> **Prior regularization and addition of noise**:
> Our Eq. 2 does not contain the KL term because it is only a loss to train the neural network to learn an approximation of the cost function. Prior regularization in our GBI posterior happens in Eq. 1, via standard Bayesian updating that balances the (generalized) likelihood and the prior probabilities. In ref 13 (Bissiri et al 2016), Eq. 7 is a loss over the entire posterior approximation, such that it can be optimized directly, thus including a prior KL term. In section 2.3, those authors note that when the generalized likelihood is defined by a cost function, the solution to the posterior approximation problem exactly follows our Eq. 1, thus the two formulations are equivalent.
>
> The addition of noise to the training samples in data space during learning of the cost function does not have any effect on regularizing the posterior density in parameter space. Instead, it expands the range of prior simulations such that the simulation support may cover observed data for which the simulator is otherwise misspecified, and does not assume local Euclidean distance.
>
> **Alternate MCMC methods**:
> Multi-chain slice-sampling was used for computational efficiency in our experiments. Any MCMC algorithm, such as Metropolis-Hastings, is applicable, since the cost estimation network simply provides the (generalized) log-likelihood. We will note this in the discussion of limitations (line 335).

---

> > ### Comment · Reviewer_CFba · 2023-08-14
> > **Response**
> >
> > Thanks for the detailed response. Please do include those clarifications in the main paper/supplementary materials.
> >
> > I will happily increase my score.

---

### Official Review · Reviewer_bf2N · 2023-07-20

**Soundness:** 3 good
**Presentation:** 3 good
**Contribution:** 3 good
**Rating:** 7
**Confidence:** 4

**Summary:**

The paper presents a new technique - amortized cost estimation (ACE) - that, as stated on the can, amortizes a broad class of loss functions used in generalized Bayesian inference (GBI) in place of the (log) likelihood.
After training on a moderate-to-large number of model simulations (10K-100K in the examples of the paper), the amortized loss can then be used in place of the real loss to perform GBI (e.g., via MCMC). Crucially, this step does not require further simulations from the model.
The authors show that the amortized loss generally matches the ground-truth loss, and show that their amortized method inherits the advantages of GBI, in that it is typically more robust to misspecification than standard and amortized Bayesian inference, as in neural likelihood estimation and neural posterior estimation

**Strengths:**

- **Quality:** The general polish and quality of the paper is high.
- **Clarity:** The paper is well written and generally clear. The aim and methodology are very well explained and motivated, with some points that would benefit from further expansion.
- **Significance:** The idea of the paper (amortizing GBI for simulator-based inference, with the additional goal of dealing with misspecification) is very timely and there are surely many applications.
- **Originality:** The idea at the core is not particularly original - amortizing the loss in GBI seems a natural direction for the field at the moment - but clearly worth pursuing. Similarly, the execution seems fairly straightforward (which is not a bad thing).


### Post-rebuttal:

Thanks for having addressed most of my comments. I appreciated the additional experiments with tempered NLE, with varying noise augmentation, and the recommendations for the choice of $\beta$. The proposed heuristic seems both reasonable and practical. I understand that given the limited time, it was not feasible to conduct an additional set of experiments.

Overall, I am satisfied with the changes and I increased my score accordingly from 6 to 7.

**Weaknesses:**

Most of my concerns have to do with the treatment of the hyperparameter $\beta$ (inverse temperature), a key element of GBI.

The role of $\beta$ could be explained and discussed more (it is mentioned in the **Limitations**).

Many experiments show the results for a range of $\beta$ (which also changes from experiment to experiment), and it is unclear how the practitioner should choose its value. For example, $\beta \rightarrow \infty$, (generalized) Bayesian inference tends to maximum-a-posteriori (or minimum-generalized-loss) point estimation. The paper is ambiguous on what the user should be trying to achieve. I appreciate that some of these issues are not of ACE (the authors' method) but of GBI, but the paper would benefit from directly addressing these points.

The comparison between GBI and Bayesian inference is somewhat unfair in that Bayesian inference is not allowed an inverse temperature hyperparameter (i.e., likelihood tempering), but in principle it could be easily applied (at least for NLE). Is ACE+GBI truly better here, or is the advantage just given by the fact that ACE/GBI has an extra free hyperparameter? In some cases, standard Bayesian inference can do better with large $\beta$. We know this can be the case for example with neural network posteriors (possibly a case of prior misspecification).
- With NLE, just run MCMC with a scaling factor $\beta$ on the amortized log likelihood.
- Amortizing different temperatures with NPE would be more complex. Naively, one would need to retrain the network for different values of $\beta$. One could also subtract the log prior from the NPE posterior to get the log likelihood, and then rescale it, but that might lead to instabilities.

Similarly to $\beta$ (but less importantly), the paper could explore more the role of the noise $\sigma$ added to $S$ observations. This parameter seems to be less relevant though - it is very nice that the authors use a fixed value throughout the paper, and this value can be determined from prior-predictive checks. Nonetheless, a bit of exploration / insight could help.

For the rest, the empirical evaluation is acceptable, but it would have been nice to see at least a couple of applications to real-world data (even another simple one), since this is arguably (also according to the authors) where GBI shines.

**Questions:**

- Can you expand on the role of $\beta$, e.g. how you chose (and a practitioner should choose) reasonable ranges of the hyperparameter?
- What about having an inverse temperature to the log likelihood of standard Bayesian inference (as per standard tempering, also seen helping in Bayesian deep learning applications)? Can you show that in your examples (for NLE, and possibly NPE)?
- Consider adding a lesion study or analysis about the role of $\sigma$ and $S$.
- Not strictly necessary, but it would be nice to see the method at work on another (simple) example with real data.

**Limitations:**

The authors address the limitations of their method (and I already asked above to expand on the role of $\beta$).
The work has no particular potential negative social impact (at least not above any other general method for statistical inference).

---

> ### Author Rebuttal · Authors · 2023-08-08
>
> We appreciate the reviewer’s encouraging remarks regarding the timeliness and broad applicability of our work in leveraging GBI for (misspecified) SBI problems, as well as noting the high quality of execution and presentation in our paper. Furthermore, they raised several interesting questions and comments regarding the effect of the beta and sigma hyperparameters in our original results, for which we conducted additional experiments and discuss below.
>
> **Experiments with tempered NLE**:
> We very much agree with the reviewer that the standard Bayesian posterior from neural SBI methods (e.g., NLE) could in theory benefit from a temperature hyperparameter, and therefore could constitute a more fair experiment vs. ACE. Therefore, we implemented “tempered NLE” by adding a similar beta parameter (inverse temperature) to the log-likelihood term for MCMC sampling, i.e., beta*NLE_loglikelihood + logprior.
>
> In short, we find that increasing beta for tempered NLE does not improve posterior predictive distance as it does for GT/ACE-GBI, and is more often detrimental. Results on all tasks are presented in Fig. 1 of the attached pdf (10k training budget). The one exception is the 2 Moons misspecified observation task, where higher beta improves NLE, but ACE still systematically outperforms tempered NLE at all betas.
>
> It’s an open question whether tempering NLE is detrimental due to targeting of entirely different objectives (likelihood vs. predictive distance), or if tempering exacerbates errors in the normalizing flow’s approximation of the likelihood. We will include the new results and discussions for potential future investigations in the paper, and we thank the reviewer for this interesting suggestion.
>
> **Experiments with varying noise augmentation (sigma)**:
> Similarly, we agree that the role of sigma could be of interest to explore further. We therefore conducted additional experiments by varying our data augmentation procedure under a simulation budget of 10k: we trained ACE with sigma of 0, 2, and 5 (original results with sigma=2), as well as entirely removing data augmentation, i.e., no noised simulations nor real observations seen during training.
>
> Overall, varying noise augmentation during training has barely any effect on predictive distance across all tasks (Fig. 2 in attached pdf). This may be due to the fact that the number of augmented samples (100) is small compared to the simulation budget (10k), providing reassurance regarding the robustness of our main results. We also studied the effect of removing training augmentation with a training budget of 1000 (not shown), which slightly decreased performance in two tasks where the observation is misspecified. This supports our original motivation for noise and real data augmentation as expanding prior simulation range to combat misspecification, though its effect is more pronounced with smaller training simulation budgets.
>
> As the reviewer pointed out, we kept sigma constant throughout our original experiments, and it is straightforward to determine a good value from prior predictive checks. These new experiments nevertheless provide interesting insights and sanity checks, which will be included in the results and discussion sections of the paper, and we thank the reviewer for their suggestions.
>
> **Discussion of beta and guideline for practitioners**:
> We agree that the non-trivial role of beta can be mentioned earlier in the work, and will discuss it in more detail in the methods section in the context of GBI. As the reviewer pointed out, while the issue is not unique to ACE but GBI in general, using our proposed method nevertheless requires the practitioner to choose a beta value suitable for their goals:
>
> Taking the log of Eq. 1, we see that beta weighs the cost function against the impact of the log-prior regularization. In this particular case, a Monte Carlo estimate of the cost function (i.e., distances) is a quantity that can be computed on simulated data alone (unlike log-likelihood in e.g., tempered NLE). As such, a practitioner should choose a value of beta that scales the cost function relative to the log prior probability, both of which can be computed on prior simulations before training ACE. In addition, the choice of beta should consider how broadly the cost function is distributed, which can be straightforwardly estimated with a 1D histogram. Since the cost is always greater or equal to 0, larger betas penalize larger costs more heavily due to the exponential.
>
> Considering the above, one reasonable heuristic is to choose beta such that it scales the mean or median of the empirical distribution of distances (computed on random pairs of prior simulations) to be in the same range as the mean or median of log-prior probabilities of those same prior samples. As a concrete example, for the Uniform 1D problem, the log-prior is -1.1 (uniform), while the median of prior simulation distances from 10k random pairs is 0.077. Therefore, a beta of 10-20 is a good start, and increasing by a factor of 5-10 further trades off posterior sample diversity for predictive simulation distance / performance.
>
> Lastly, many GBI methods for setting beta have been developed (see, e.g., Wu, Martin, Bayesian Analysis 2023). Similar methods could be applied to ACE and, since ACE amortizes inference over beta, it is even amenable to methods which require repeated posterior sampling for different values of beta.
>
> Thanks to the reviewer for raising this important practical issue, and we will include a summary of the above in the method and discussion sections.
>
> **Another application with real data**:
> Given the time constraint of the rebuttal phase, we are not able to conduct an additional set of experiments on a real problem, which requires a suitable dataset and simulator model, in-house expertise, and comparison with existing methods. However, we agree that further applications on real problems are important, and are currently being pursued.

---

> > ### Comment · Reviewer_bf2N · 2023-08-10
> > **Response to the rebuttal**
> >
> > Thanks for having addressed most of my comments. I appreciated the additional experiments with tempered NLE, with varying noise augmentation, and the recommendations for the choice of $\beta$. The proposed heuristic seems both reasonable and practical.
> > I understand that given the limited time, it was not feasible to conduct an additional set of experiments.
> >
> > Overall, I am satisfied with the changes and I will argue for acceptance of the paper. I will increase my score accordingly.

---

> > > ### Author Response · Authors · 2023-08-14
> > > **Question regarding score increase**
> > >
> > > Thanks again for the feedback and we are happy to hear that the reviewer will increase the score and argue for acceptance!
> > >
> > > Unfortunately, the score has not been changed on OpenReview yet, so we would kindly like to ask if you simply forgot to increase the score, if you need any further clarification on the paper, or if this is a technical issue with OpenReview. Thank you very much for your time!

---

> > > > ### Comment · Reviewer_bf2N · 2023-08-14
> > > > **Score increased**
> > > >
> > > > Done. For some reason, the edit button on the original review was not available at the time, but it is now. I increased the score from 6 to 7.

---

### Author Rebuttal · Authors · 2023-08-08

We would like to thank all the reviewers for their constructive and detailed engagement with our work, resulting in many helpful comments, questions, and opportunities for clarification, as well as ideas for future work. We are especially grateful for several reviewers’ acknowledgement that the problem we tackle is “well-motivated” (bf2N), “important and relevant” (HKbp), that our contribution to the SBI & GBI literature is novel (CFba, HKbp, 9aMX), “timely” (bf2N), and will be of value to the community (CFba, HKbp, 9aMX), while also pointing out that the method itself being simple is a good thing (bf2N, HKbp). We are happy to hear that all 4 reviewers thought the paper was “well-written” and clearly structured, and appreciated the open-source code (HKbP, 9aMX).

At the same time, we agree with many of the concerns and questions raised by the reviewers, and have responded individually to each review in detail. We summarize here the main points and results from new experiments, referring to individual reviewers’ comments whenever applicable. We hope that these follow-up discussions and results clarify their concerns, allowing all reviewers to recommend our work for acceptance.

**Tempered NLE (rev:bf2N)**:
We implemented the tempered Bayesian posterior with NLE (neural likelihood estimation), and tested it on all tasks with the same betas as used for GBI. Overall, we find that tempering NLE does not result in lower posterior predictive distances that we aim to minimize, and is in fact often detrimental compared to standard NLE. “Colder” posteriors (higher beta) slightly improved NLE in just one of the eight scenarios (2 Moons, misspecified observations), but is still outperformed by ACE in this (and all other) tasks. Results in attached pdf, Fig. 1. We will update Fig. 3, method and discussion sections in the paper.

**Noise augmentation lesion study (rev:bf2N)**:
We varied the augmentation noise variance (sigma=0,2,5) during training for ACE in the 10k simulation budget experiment and observed no impact in subsequent performance. We further removed all training data augmentation, including observed data points, and saw barely any effects (attached pdf, Fig. 2). We therefore conclude that at sufficient simulation budgets, removing augmentation would not degrade performance. We did see a slight decrease in performance for two tasks with misspecified observations when all augmentation was removed for ACE trained with 1k simulation budget, supporting the idea that data augmentation aids learning outside of regions produced by the simulator. These results will be discussed in the paper.

**On choosing beta (rev:bf2N) and when beta approaches infinity (rev:bf2N & CFba)**:
We agree that earlier acknowledgement of the additional hyperparameter increases transparency, and will do so in the method section as well. We also agree that choosing beta is an important practical concern, and provide here a heuristic for choosing starting values: a good “baseline value” is such that it scales the average distance across a subset of the training data (precomputed on prior simulations) to be in the same range as the (log) prior probability, both of which can be computed on prior simulations. From there, increasing beta sacrifices sample diversity for predictive distance, as was pointed out that when beta approaches infinity, posterior samples converge onto the minimizer of the cost function. We also note that, since our method is amortized over beta, beta-selection methods which require posterior sampling for several different values can be performed at low computational cost. See individual responses for more details.

**Prior regularization via dKL (rev:CFba)**:
In ref 13, the GBI loss is for optimizing the entire posterior approximation directly, hence prior regularization is included explicitly as the KL term. In contrast, our loss function (Eq. 2) is only for training the cost estimation network, while prior regularization occurs via the standard Bayesian updating in Eq. 1. These two views are consistent, as is also stated in ref 13, Sec. 2.3.

**Clarification on experimental details and results (rev:CFba)**:
We apologize for erroneous or missing references to methodological details and definitions. We now refer to the definition of well-specified and misspecified observations in the Appendix (A4.3), fixed inconsistencies when referring to distance vs. cost functions and how ground-truth was obtained, and make further clarifications regarding the MSE vs. MMD, and their differing directions of change compared to C2ST with increasing beta. Full details in the individual response.

**Relationship to uncertainty estimation and Bayesian NN (rev:HKbp)**:
We discuss how posterior predictive samples and their distance to the observation (our optimized quantity) conceptually relate to learning predictive uncertainty in works on uncertainty estimation (e.g., epistemic neural networks) and Bayesian NNs, while noting the very different goals and implementations between these two areas of literature.

**Acknowledgement of limitations regarding finite data, high-dimensional problems, hyperparameter sensitivities**:
We thank the reviewers for noting further limitations in our work that we had failed to discuss, and will explicitly acknowledge in the discussion section, including: errors in neural network approximation when learning from finite data (rev:HKbp), potential issues and strategies when applying to higher dimensional problems (rev:HKbp), and sensitivity to various hyperparameters,e.g., beta, sigma, and training budget. Detailed discussions in individual responses.

**Clarification and improvements on presentation and visualization**:
We make a number of clarifications or changes following questions and suggestions raised by all reviewers, including adjustment to Fig. 2C (rev:9qMX), moving the propositions to Appendix for clarity (rev:9qMX), and further explanations on results in Fig. 3 (rev:CFba).

---

### Comment · Area_Chair_vU7t · 2023-08-15
**Author-reviewer discussion**

Dear all,

The author-reviewer discussion period has now started. It will continue for one more week, until August 21.

@authors: Please respond to the comments or questions reviewers may further have. Remain short and to the point.

@reviewers: Please read the author's responses and ask any further questions you may have. To facilitate the decision by the end of the process, please also acknowledge that you have read the responses and indicate whether you want to update your evaluation.

- You can update your evaluation positively (if you are satisfied with the responses) or negatively (if you are not satisfied with the responses or share other reviewers' concerns). Please note that major changes are a reason for rejection.
- You can also keep your evaluation unchanged. In this case, please indicate that you have read the responses and that you do not have any further comments.

Best regards,
The AC

---

### Decision · Program_Chairs · 2023-09-21

**Decision:**

Accept (poster)

**Comment:**

The reviewers unanimously recommend acceptance (7-6-6-7). Several minor issues have been raised and discussed during the author-reviewer discussion period. The authors are encouraged to take this feedback into account in the final version of the paper.